# Pyridylpiperazine efflux pump inhibitor boosts in vivo antibiotic efficacy against *K. pneumoniae*

Anais Vieira Da Cruz[1,6], Juan-Carlos Jiménez-Castellanos[2,6], Clara Börnsen [3,6], Laurye Van Maele [2,6], Nina Compagne[1], Elizabeth Pradel [2], Reinke T Müller[4], Virginie Meurillon[1], Daphnée Soulard[2], Catherine Piveteau[1], Alexandre Biela [1], Julie Dumont[1], Florence Leroux [1,5], Benoit Deprez[1], Nicolas Willand[1], Klaas M Pos [4,7✉], Achilleas S Frangakis [3,7✉], Ruben C Hartkoorn [2,7✉] & Marion Flipo [1,7✉]

## Abstract

**Antimicrobial resistance is a global problem, rendering conventional treatments less effective and requiring innovative strategies to combat this growing threat. The tripartite AcrAB-TolC efflux pump is the dominant constitutive system by which Enterobacterales like *Escherichia coli* and *Klebsiella pneumoniae* extrude antibiotics. Here, we describe the medicinal chemistry development and drug-like properties of BDM91288, a pyridylpiperazine-based AcrB efflux pump inhibitor. In vitro evaluation of BDM91288 confirmed it to potentiate the activity of a panel of antibiotics against *K. pneumoniae* as well as revert clinically relevant antibiotic resistance mediated by *acrAB-tolC* overexpression. Using cryo-EM, BDM91288 binding to the transmembrane region of *K. pneumoniae* AcrB was confirmed, further validating the mechanism of action of this inhibitor. Finally, proof of concept studies demonstrated that oral administration of BDM91288 significantly potentiated the in vivo efficacy of levofloxacin treatment in a murine model of *K. pneumoniae* lung infection.**

**Keywords** AcrAB-TolC; Antimicrobial Resistance; Cryo-EM; Efflux Pump Inhibitor; Antibiotic Efflux Pump
**Subject Categories** Microbiology, Virology & Host Pathogen Interaction; Pharmacology & Drug Discovery; Structural Biology

## Introduction

Infections caused by multidrug-resistant (MDR) Gram-negative bacteria are a major concern for human healthcare (WHO Antimicrobial resistance, 2021). In 2019, an estimated 4.95 million deaths were associated with bacterial antimicrobial resistance (AMR), with *Escherichia coli* and *Klebsiella pneumoniae* being the two leading pathogens (Murray et al, 2022). Various mechanisms, such as the expression of β-lactamases or point mutations in targeted proteins, impart strong antibiotic-specific resistance, while the overproduction of efflux pumps can confer broad-spectrum antibiotic resistance (Li et al, 2015; Webber, 2003). While physiological expression of efflux pumps leads to basal compound extrusion and decreases antibiotic susceptibility, constitutive efflux pump overexpression, as a consequence of loss-of-function mutations in transcriptional regulators, has been linked to clinically relevant antibiotic resistance (Webber et al, 2005; Schneiders et al, 2003). In Gram-negative bacteria, efflux pumps belonging to the Resistance-Nodulation-cell Division (RND) superfamily play a central role in the extrusion of a myriad of chemically unrelated compounds, including fatty acids, antiseptics, detergents, virulence factors, toxins and most notably, antibiotics (Ni et al, 2020; Teelucksingh et al, 2022). *K. pneumoniae* and *E. coli* encode several RND-pumps, but in both species AcrAB-TolC is the major constitutive tripartite efflux pump system and their AcrB proteins share 92% identical residues (Padilla et al, 2010; Blair et al, 2015; Wang et al, 2017; Okusu et al, 1996; Iyer et al, 2019). AcrAB-TolC is composed of (i) an inner membrane trimeric H$^+$/drug antiporter (i.e., AcrB), (ii) a hexameric periplasmic adapter protein (i.e., AcrA), that interacts with AcrB and connects to (iii) TolC, a trimeric outer membrane channel. Via tip-to-tip interactions AcrA and TolC form a long channel guiding the drugs expelled by AcrB across the outer membrane. AcrB sequesters and transports drug substrates via a concerted conformational cycling of each of its protomers, which adopt the Loose (L), Tight (T) or Open (O) state (Murakami et al, 2006; Seeger et al, 2006; Sennhauser et al, 2006).

Constitutive basal expression of AcrAB-TolC provides *K. pneumoniae* an inherent mechanism to extrude numerous antibiotics (Jiménez-Castellanos et al, 2016; Rosenblum et al, 2011; De Majumdar et al, 2015). Current antibiotic treatment for drug

[1]Univ. Lille, Inserm, Institut Pasteur de Lille, U1177 - Drugs and Molecules for Living Systems, F-59000 Lille, France. [2]Univ. Lille, CNRS, Inserm, CHU Lille, Institut Pasteur de Lille, U1019 - UMR 9017 - CIIL - Center for Infection and Immunity of Lille, F-59000 Lille, France. [3]Buchmann Institute for Molecular Life Sciences and Institute for Biophysics, Goethe University Frankfurt, Max-von-Laue-Str. 15, D-60438 Frankfurt am Main, Germany. [4]Institute of Biochemistry, Goethe-University Frankfurt, Max-von-Laue-Str. 9, D-60438 Frankfurt am Main, Germany. [5]Univ. Lille, CNRS, Inserm, CHU Lille, Institut Pasteur Lille, US 41—UAR 2014—PLBS, F-59000 Lille, France. [6]These authors contributed equally as first authors: Anais Vieira Da Cruz, Juan-Carlos Jiménez-Castellanos, Clara Börnsen, Laurye Van Maele. [7]These authors contributed equally: Klaas M Pos, Achilleas S Frangakis, Ruben Hartkoorn, Marion Flipo. ✉E-mail: pos@em.uni-frankfurt.de; achilleas.frangakis@biophysik.org; ruben.hartkoorn@inserm.fr; marion.flipo@univ-lille.fr

susceptible *K. pneumoniae* however is designed to cope with this basal permeability barrier, either by not being a pump substrate, or alternatively by being sufficiently potent to allow target engagement at low intracellular concentration. Nonetheless, the potency of this latter group of antibiotics can be affected by mutations that cause overexpression of *acrAB-tolC* and thus limit intracellular drug concentration. In *K. pneumoniae*, the primary genetic determinant leading to *acrAB* upregulation is loss-of-function mutations in *ramR* that encodes a transcriptional repressor whose malfunction triggers the production of the transcriptional activator RamA, which in turn mediates the overexpression of *acrAB* (De Majumdar et al, 2015; Belmar Campos et al, 2017; Jiménez-Castellanos et al, 2018). The role of *ramR* mutations in antibiotic resistance was originally reported for tigecycline (Mao et al, 2020). However, other works found *ramA* overexpressing clinical strains to already be present in the nosocomial *K. pneumoniae* population pre-dating treatment by tigecycline (Rosenblum et al, 2011; Hentschke et al, 2010). Similarly, analysis of clinical *K. pneumoniae* isolates by Wang and colleagues (Wang et al, 2015) found 33/106 strains to carry mutations in *ramR*, correlating with not only resistance to tigecycline and minocycline but also intermediate resistance to ciprofloxacin and cefoxitin. While the contribution of efflux pumps overexpression on antibiotic resistance via *ramR* mutations remains scattered, the clinical impact of these mutations is well established (Belmar Campos et al, 2017; Zhang et al, 2021; Ruzin et al, 2008; Bratu et al, 2009). Furthermore, it also remains unclear whether efflux pump inhibitors can effectively overcome resistance mediated by such efflux pump overproduction.

In this context, the development of drug-like efflux pumps inhibitors (EPIs) appears an attractive solution for re-sensitizing bacteria toward antibiotics, potentially extending the lifespan and scope of current antibiotic therapies (Compagne et al, 2023; Douafer et al, 2019; Pagès and Amaral, 2009; Venter et al, 2015). To date, several classes of EPIs have been described to inhibit AcrAB-TolC activity and enhance antibiotic potency, such as the peptidomimetic PAßN (Lomovskaya et al, 2001), naphtylmethyl-piperazine (NMP) (Bohnert and Kern, 2005), 2H-benzo[h]chromene derivatives (WK2) (Wang et al, 2021), pyridopyrimidines (Nakashima et al, 2013) and pyranopyridines of the MBX series (Opperman et al, 2014; Nguyen et al, 2015). Despite this, there is limited data available on the in vivo effectiveness of these EPIs, and so far, none have progressed to clinical development against Enterobacterales. This may be attributed to various factors, including unfavourable pharmacokinetic parameters and toxicity issues (e.g. PAβN (Lomovskaya and Bostian, 2006)), off-target activity (e.g. NMP (Bolla et al, 2011)) or limited in vivo efficacy (e.g. MBX compounds (Opperman et al, 2018)).

We have recently described the discovery of a novel chemical family of pyridylpiperazine (PyrPip)-based allosteric inhibitors of the AcrAB-TolC efflux pump (Plé et al, 2022). In contrast to the above-mentioned EPIs, PyrPips bind to a distinct and previously unexploited allosteric pocket in the transmembrane domain of the AcrB L protomer, thereby blocking the functional catalytic cycle of the pump. Initial medicinal chemistry optimisation identified an improved PyrPip EPI, BDM88855 (**1**), with a quinoline core, capable of boosting the activity of several antibiotics on *E. coli* (Plé et al, 2022). The PyrPip scaffold has the advantage of being chemically tractable and amenable for further multi-parametric

optimisation needed to improve potency, pharmacokinetic properties, and overcome potential toxicity issues.

In this work, we sought to optimise the physico-chemical and pharmacokinetic properties of the PyrPip series of EPIs to identify a compound with favourable properties for in vivo proof-of-concept efficacy studies on *K. pneumoniae*. Microbiological characterisation and structural biology analysis using cryo-EM validated this compound to be a *bona fide* inhibitor of AcrB. Checkerboard assays confirmed its ability to revert clinically relevant antibiotic resistance mediated by constitutive efflux pump overexpression. Finally, using a murine model of a pulmonary infection by *K. pneumoniae*, we demonstrated that oral administration of this PyrPip significantly potentiated the in vivo activity of levofloxacin.

# Results

## Medicinal chemistry

With the aim of developing a PyrPip for in vivo efficacy studies in mice, it was important to optimise the metabolic stability of these efflux pump inhibitors to guarantee sufficient exposure in mice needed for antibiotic boosting. Indeed, in vitro mouse liver microsome stability studies of the originally described lead PyrPip, BDM88855 (**1**), found this inhibitor to have intermediate intrinsic clearance ($CL_{int}$ = 59 μL/min/mg proteins) which was considered suboptimal for in vivo studies (Table 1). For this reason, we sought to further explore the structure–activity relationships of PyrPips to identify inhibitors with improved metabolic stability ($CL_{int}$ < 20 μL/min/mg proteins). Given the high similarity between the *E. coli* and *K. pneumoniae* AcrB proteins, the newly synthesised PyrPip analogues were initially evaluated on *E. coli* for their ability to boost pyridomycin antibiotic activity (as in Plé et al, 2022). The microsomal stability of the most potent compounds ($EC_{90}$ < 4 μM) was then measured. Previous studies indicated that the piperazine and chlorine atom found respectively at position 2 and 3 of the quinoline ring were necessary for antibiotic boosting activity in *E. coli* (Plé et al, 2022). The co-crystal structure of BDM88855 (**1**) with *E. coli* AcrB (pdb 7OUK) confirmed this by showing that the basic nitrogen of the piperazine ring interacted with D408 via a salt bridge, while the chlorine atom formed a halogen bond with the carbonyl backbone of K940. To identify metabolically stabilised PyrPips while maintaining AcrB inhibition, it was therefore decided to explore modifications to the quinoline core as well as the introduction of substituents at position 6 of the quinoline ring.

We first changed the position of the nitrogen atom in the quinoline core and replaced this aromatic ring with a quinoxaline or a pyrrolo[2,3-b]pyridine (Table 1). The modification of the position of the nitrogen atom (compound **2**) led to a 4-fold decrease in potency ($EC_{90}$ = 15.6 μM), while the introduction of a second nitrogen atom in position 3 of the quinoline (quinoxaline core, compound **3**) was better tolerated and led to a compound with similar potency to BDM88855.HCl (**1'**) ($EC_{90}$ = 3.9 μM) but with inferior metabolic stability ($CL_{int}$ = 111 μL/min/mg proteins). Finally, the replacement of the quinoline ring with a pyrrolo[2,3-b]pyridine (compound **4**) led to a complete loss of activity ($EC_{90}$ > 250 μM).

**Table 1. Panel of synthesised PyrPip EPIs (compounds 1′–13) and their potency at boosting pyridomycin activity in *E. coli*, as well as their mouse microsomal stability.**

Structures

| Compound | X | Y | R substituent | PyrPip antibiotic boosting activity EC$_{90}$ ($\mu$M)[a] | Microsomal stability[b] | |
|---|---|---|---|---|---|---|
| | | | | | t$_{1/2}$ (min) | CL$_{int}$ ($\mu$L/min/mg proteins)[c] |
| 1′ (BDM88855.HCl) | CH | N | H | 3.6 ± 0.5 | 24 | 59 |
| 2 | N | CH | H | 15.6 | ND | ND |
| 3 | N | N | H | 3.9 ± 1.6 | 15 | 111 |
| 4 | | |  | > 250 | ND | ND |
| 5 (BDM91288) | CH | N |  | 3.3 ± 1.7 | > 40 | 6 |
| 6 | CH | N |  | 104 ± 28 | 11 | 103 |
| 7 | CH | N |  | 13 ± 4 | 6 | 223 |
| 8 | CH | N |  | 31 | ND | ND |
| 9 | CH | N |  | 3.9 | 40 | 47 |
| 10 | CH | N |  | 3.9 | > 40 | 2 |
| 11 | CH | N |  | 7.8 | ND | ND |
| 12 | CH | N |  | 1.3 ± 0.4 | 2.6 | 303 |
| 13 | CH | N |  | 9.1 ± 4.3 | ND | ND |

*ND* not determined.
[a]EC$_{90}$ represents the 90% maximal effective concentration of tested compounds that prevents the growth of *E. coli* BW25113 in the presence of 8 $\mu$g/mL pyridomycin as measured by resazurin reduction (the MIC$_{90}$ of pyridomycin alone is 12.5–25 $\mu$g/mL). Data are the result of at least two biological replicates and are presented as mean values ± SEM.
[b]Metabolic stability in mouse liver microsomes.
[c]CL$_{int}$ = intrinsic clearance.

Table 2. Modulation of antibiotic activity by EPIs BDM88855.HCl (1'), BDM91288 (5) and PAβN in *K. pneumoniae* strains.

| Antibiotics | *K. pneumoniae* WT | | | | *K. pneumoniae* AcrB^V448E | | | | *K. pneumoniae* RamR^mut | | | | *K. pneumoniae* AcrB^V448E RamR^mut | | | |
|---|---|---|---|---|---|---|---|---|---|---|---|---|---|---|---|---|
| | No EPI | 1' | 5 | PAβN | No EPI | 1' | 5 | PAβN | No EPI | 1' | 5 | PAβN | No EPI | 1' | 5 | PAβN |
| Levofloxacin | 0.04 | *0.01#* | *0.01#* | *0.02#* | 0.04 | 0.03 | 0.03 | *0.02#* | **0.64** | *0.08#* | *0.05#* | *0.04#* | **1.28** | 1.28 | 1.28 | *0.05#* |
| Azithromycin | 16 | *2.7#* | *2.0#* | *0.5#* | 21 | *6.7#* | *11#* | *0.5#* | **37** | *11#* | *3.3#* | *0.7#* | 21 | 37 | 21 | *0.7#* |
| Linezolid | 128 | *13#* | *27#* | *64#* | 128 | 85 | 128 | *64#* | 128 | 128 | 107 | 128 | 128 | 128 | 128 | 128 |
| Chloramphenicol | 4.0 | *1.0#* | *1.0#* | *1.7#* | 6.7 | *2.0#* | 4.7 | *2.0#* | **53** | *2.7#* | *4.7#* | *2.7#* | **128** | 128 | 128 | *5.3#* |
| Novobiocin | 64 | *13#* | *13#* | *11#* | 128 | *64#* | *32#* | *16#* | 128 | 128 | 85 | *43#* | 128 | 128 | 128 | *8#* |
| Oxacillin | 128 | *32#* | *43#* | 128 | 128 | 128 | 128 | 128 | 128 | 128 | 128 | 128 | 128 | 128 | 128 | 128 |
| Tetracycline | 0.8 | *0.4#* | *0.4#* | 1.3 | 1.9 | *0.8#* | *0.7#* | 2.7 | **8.5** | *1.1#* | *0.7#* | *4.3#* | **11** | 11 | 8.5 | *4.3#* |
| Cefepime | 0.09 | 0.05 | 0.05 | 0.21 | 0.06 | 0.05 | 0.04 | 0.64 | **0.53** | 0.64 | *0.11#* | 0.64 | **0.75** | 0.85 | 0.96 | 0.85 |
| Piperacillin | 8.0 | 5.3 | *4.0#* | 8.0 | 16 | *5.3#* | *5.3#* | 13 | **27** | 16 | *6.7#* | 21 | **21** | 37 | 21 | 43 |
| Fusidic acid | 128 | 85 | 128 | *21#* | 128 | 128 | 128 | *32#* | 128 | 128 | 128 | 85 | 128 | 128 | 128 | 107 |
| Ceftazidime | 0.43 | 0.32 | 0.53 | 0.32 | 0.85 | *0.37#* | *0.32#* | 0.53 | **1.3** | 1.1 | *0.64#* | *0.64#* | **1.28** | 1.28 | 1.28 | 1.28 |
| Aztreonam | 0.07 | 0.11 | 0.11 | 0.16 | 0.13 | 0.16 | 0.21 | 0.32 | **0.43** | 0.32 | 0.27 | 0.53 | **0.53** | 0.53 | 0.75 | 0.53 |
| Meropenem | 0.50 | 0.50 | 0.50 | 0.50 | 0.50 | 0.67 | 0.50 | 0.50 | 0.50 | 0.50 | 0.50 | 0.50 | 0.67 | 0.50 | 0.50 | 1.0 |

Minimum Inhibitory Concentration (mg/L) for a panel of antibiotics in the presence and absence of EPI (100 μM BDM88855.HCl (1'), BDM91288 (5) and PAβN) against *K. pneumoniae* ATCC 43816 (WT), and isogenic mutants bearing a point mutation in *acrB* (AcrB^V448E), *ramR* (RamR^mut = K124*stop) or both (AcrB^V448E, RamR^mut). Bold numbers indicate basal antibiotic MICs that are shifted in the *K. pneumoniae* mutants compared to WT. Conditions where EPIs decrease the antibiotic MIC by at least two-fold are indicated by #. Values are the mean MICs of at least three biological replicates.

As the co-crystal structure of BDM88855 (1) with AcrB (pdb 7OUK) shows that there is significant space for PyrPip expansion at the C-6 position of the quinoline core, we decided to introduce at this position substituents containing an amine that could potentially interact with acidic residues present at the cytoplasmic side of the AcrB binding pocket (i.e., E947/D951 in *E. coli* or E946/D950 in *K. pneumoniae*). It was observed that the introduction of a piperazine was tolerated (compound 5, $EC_{90}$ = 3.3 μM), and this modification led to improved metabolic stability in mouse liver microsomes ($CL_{int}$ = 6 μL/min/mg proteins). We then investigated if the basic nitrogen atom of this second piperazine was necessary for the antibiotic boosting activity, and found that the morpholine (compound 6, $EC_{90}$ = 104 μM, $CL_{int}$ = 103 μL/min/mg proteins) and piperidine (compound 7, $EC_{90}$ = 13 μM, $CL_{int}$ = 223 μL/min/mg proteins) analogues showed a decrease in potency and microsomal stability. The inferred advantage of the basic nitrogen in the second piperazine of compound 5 suggested a possible interaction with the aforementioned acidic residues of AcrB. Substitution of the piperazine with a piperazinone (compound 8, $EC_{90}$ = 31 μM) led to an 8-fold decrease in potency. Piperazine ring opening (compound 9, $EC_{90}$ = 3.9 μM) maintained potency but decreased the compound microsomal stability ($CL_{int}$ = 47 μL/min/mg proteins), while piperazinone ring opening (compound 10, $EC_{90}$ = 3.9 μM) was tolerated and allowed for good microsomal stability ($CL_{int}$ = 2 μL/min/mg proteins). Increasing the chain length of compound 9 by adding a methylene group led to a 2-fold decrease in potency (compound 11, $EC_{90}$ = 7.8 μM). The introduction of 3-(R)-aminopyrrolidine (compound 12, $EC_{90}$ = 1.3 μM) led to a 3-fold improvement of potency compared to analogue 5, but this compound was metabolically instable ($CL_{int}$ = 303 μL/min/mg proteins), while the (S)-enantiomer (compounds 13, $EC_{90}$ = 9.1 μM) was less active. None of the compounds showed significant antibacterial activity against *E. coli* by themselves ($MIC_{90}$ > 250 μM). Based on these results, compound 5 showed the optimal compromise between potency and microsomal stability and was selected as a promising candidate for further in vitro and in vivo evaluation.

## PyrPips target AcrB in *K. pneumoniae*

To evaluate whether PyrPip efflux pump inhibition observed on *E. coli* AcrB translates to *K. pneumoniae* AcrB, compounds BDM88855.HCl (1') and BDM91288 (5) were tested with a panel of known AcrAB-TolC substrate antibiotics on *K. pneumoniae* (Table 2). To confirm that the observed boosting of antibiotic activity was through AcrB inhibition, PyrPip-resistant *K. pneumoniae* mutants were then isolated on solid media containing both BDM88855.HCl (1') [500 μM]$_F$ and a sub-inhibitory concentration of erythromycin (50 μg/mL). Unlike with *E. coli* (Plé et al, 2022), no isolate could be obtained that was purely resistant to PyrPip without altered antibiotic susceptibility. Indeed, all *K. pneumoniae* clones isolated were resistant to both BDM88855.HCl (1') and erythromycin itself (as well as to levofloxacin, suggesting a general mechanism of resistance). Whole-genome sequencing of one such resistant isolate found it to carry two non-synonymous mutations: one in *acrB* (t1342a, [V448E]), and the other in *ramR* (a370t, [K124*]). The identified substitution in AcrB was located between the two residues where mutations were previously identified to confer PyrPip resistance in *E. coli* (Plé et al, 2022), strongly suggesting its involvement in the BDM88855.HCl (1') resistance phenotype. Regarding the null mutation in *ramR*, such variants are well documented to cause overexpression of *acrAB* through dysregulation of the transcriptional activator RamA. This mediates antibiotic resistance to pump substrates such as tigecycline, cephalosporins, tetracycline, and fluoroquinolones (De Majumdar et al, 2015; Jiménez-Castellanos et al, 2018; Wang et al, 2015), and is likely responsible for the increased erythromycin and levofloxacin resistance level. To

separate the role of the AcrB mutation (V448E) and the RamR-null mutation in the observed resistance, isogenic strains carrying either mutation were constructed (*K. pneumoniae*-AcrB$^{V448E}$ and *K. pneumoniae*-RamR$^{mut}$ respectively). *K. pneumoniae*-AcrB$^{V448E}$ showed unaltered basal antibiotic susceptibility, but was resistant to antibiotic boosting by BDM88855.HCl (**1'**) or BDM91288 (**5**), while it remained sensitive to antibiotic boosting by PAβN (Table 2), validating that in *K. pneumoniae* the PypPips inhibit AcrB in a similar manner to that described in *E. coli*. In contrast, *K. pneumoniae*-RamR$^{mut}$ was confirmed to be resistant to several antibiotics known to be AcrAB-TolC substrates (Table 2) while BDM88855.HCl (**1'**) and BDM91288 (**5**) mediated antibiotic boosting was maintained.

## Direct inhibition of *K. pneumoniae* AcrB by BDM91288 (5)

To confirm the direct inhibition of AcrB by PyrPips, we assessed the impact of BDM91288 (**5**) on the accumulation of the AcrB substrate berberine in *K. pneumoniae* using a fluorescence uptake assay. As expected, BDM91288 (**5**) mediated a concentration-dependent stimulation of berberine uptake in *K. pneumoniae* WT with an apparent Ki of around 20 µM, which was not observed for *K. pneumoniae* producing the AcrB variant V448E (Fig. EV1).

## Localization of BDM91288 (5) binding site in AcrB by cryo-EM

To further validate that AcrB is the target of PyrPips in *K. pneumoniae*, we determined the co-structure of *K. pneumoniae* AcrB with BDM91288 (**5**) at 2.97 Å resolution via single-particle cryo-EM (Figs. 1A and EV2; Appendix Fig. S1; Appendix Tables S1 and S2). Non-proteinaceous electron density between transmembrane (TM) segments TM4, TM5 and TM10 showed BDM91288 (**5**) to bind to the TM domain of the L protomer (Fig. 1B) in a similar manner to that described for BDM88855 (**1**) with *E. coli* AcrB (Plé et al, 2022). Together this corroborated the specificity of PyrPips for the AcrB L protomer, the congruent binding position within the TM domain and confirmed the salt bridge formation between the -NH$_2^+$- group of the protonated piperazine moiety with the charged carboxylic side chain of D408 (Fig. 1D). Additionally, interactions of BDM91288 (**5**) with other residues of the proton relay network (D407, D408, and K939), L449, and the carbonyl oxygen of the main chain L404 (Fig. EV2) were observed. Relative to BDM88855 (**1**), the second piperazine introduced in BDM91288 (**5**) was found to interact with the acidic side chain of D950 at the cytoplasmic entrance of the L protomer (Fig. 1C). The overall interaction of BDM91288 (**5**) revealed therefore two salt bridges, one with D408 involving the protonated secondary nitrogen atom (-NH$_2^+$-) of the first piperazine moiety, and the other with D950 involving the -NH$_2^+$- of the second piperazine moiety, as expected from the medicinal chemistry design. However, this additional salt bridge with D950 did not lead to an increase in potency. BDM91288 (**5**) is attached to the binding site via a directional 2-point suspension, which likely locks the protomer in a state that can be described as an intermediate conformation between the O and L state. The locked O/L intermediate conformation is anticipated to result in the inhibition of the interprotomer-dependent functional cycling of the entire AcrB trimer, thereby preventing antibiotic efflux.

## EPI potency and reversal of RamR-mediated levofloxacin resistance

Checkerboard assays were performed to determine the concentration-dependent EPI potency of BDM91288 (**5**) and PAβN on levofloxacin activity. For the parental *K. pneumoniae* strain, BDM91288 (**5**) readily boosted levofloxacin activity starting at around 9 µM, with a maximal 4-fold boosting observed at 38 µM (Fig. 2), with similar boosting observed for PAβN. In the case of *K. pneumoniae* AcrB$^{V448E}$, boosting was no longer observed for BDM91288 (**5**) while it was maintained by PAβN (Fig. 2). For *K. pneumoniae* RamR$^{mut}$, levofloxacin MIC was increased from 62.5 to 500 ng/mL, and this resistance could be completely reverted by BDM91288 (**5**) to reach the same boosted MIC as observed in *K. pneumoniae* WT (16 ng/mL). Additionally, it could only be partially reverted by PAβN (Fig. 2). However, it needs to be noted that higher PyrPip and PAβN concentrations were required to achieve full levofloxacin sensitisation, likely due to the increased AcrB expression (Fig. 2). No modulation of levofloxacin MIC was observed by BDM91288 (**5**) in the selected *K. pneumoniae* AcrB$^{V448E}$RamR$^{mut}$ mutant confirming that the BDM91288 (**5**) mediated reversion of resistance was through AcrB (Fig. 2). Overall, our data clearly demonstrates that PyrPips boost antibiotic activity in *K. pneumoniae* through the inhibition of AcrB, furthermore, BDM91288 (**5**) can fully revert levofloxacin resistance even in a *ramR* null mutant in vitro.

## Pharmacokinetics of BDM91288 (5) in mice

In addition to effective antibiotic boosting activity and suitable microsomal stability, BDM91288 (**5**) showed high solubility (≥1 mM in PBS), low plasma protein binding (F$_{unbound}$ = 54%), weak inhibition of the hERG potassium channel (8% at 30 µM) and weak in vitro cytotoxicity on BALB/3T3 cells (CC$_{50}$ = 48 µM). Furthermore, no toxicity was observed in mice over a 24-h period after a single oral administration of BDM91288 (**5**) at 30 mg/kg. For these reasons, this compound was selected for multi-organ pharmacokinetic (PK) studies to measure its temporal distribution and tissue concentration in female mice (CD-1), following a single oral administration of 30 mg/kg (Fig. EV3A; Appendix Table S3). The data showed BDM91288 (**5**) to be bioavailable with detectable concentrations in the plasma (C$_{max}$ = 2.0 µM, T$_{max}$ = 4 h, area under the plasma drug concentration-time curve (AUC$_{(0-24h)}$) = 27 h.µM), though below the concentration needed for efficacy in vitro on *K. pneumoniae* WT (9–38 µM for boosting levofloxacin activity in vitro, Fig. 2). In contrast, BDM91288 (**5**) PK parameters measured in the whole lung showed significant accumulation in this organ (C$_{max}$ = 225 µM, T$_{max}$ = 4 h, AUC$_{(0-24h)}$ = 3891 h.µM), with concentrations well above the in vitro effective concentration (Figs. 2 and EV3A). PK studies in female mice (CD-1) following administration of a single BDM91288 (**5**) dose (30 mg/kg, *per os* (p.o.)) with a single levofloxacin dose (10 mg/kg, intraperitoneal (i.p.)) were performed to measure the concentration in the epithelial lining fluid (ELF) at 3 time points (2 h, 6 h, and 24 h) (Fig. EV3B; Appendix Table S4). Analysis of the ELF showed a BDM91288 (**5**) exposure (AUC$_{(0-24h}$ = 62 h.µM) 2.3-fold higher than that measured in the plasma, but 1.6% of that measured in the whole lung. The accumulation of BDM91288 (**5**) in the lung may be due to

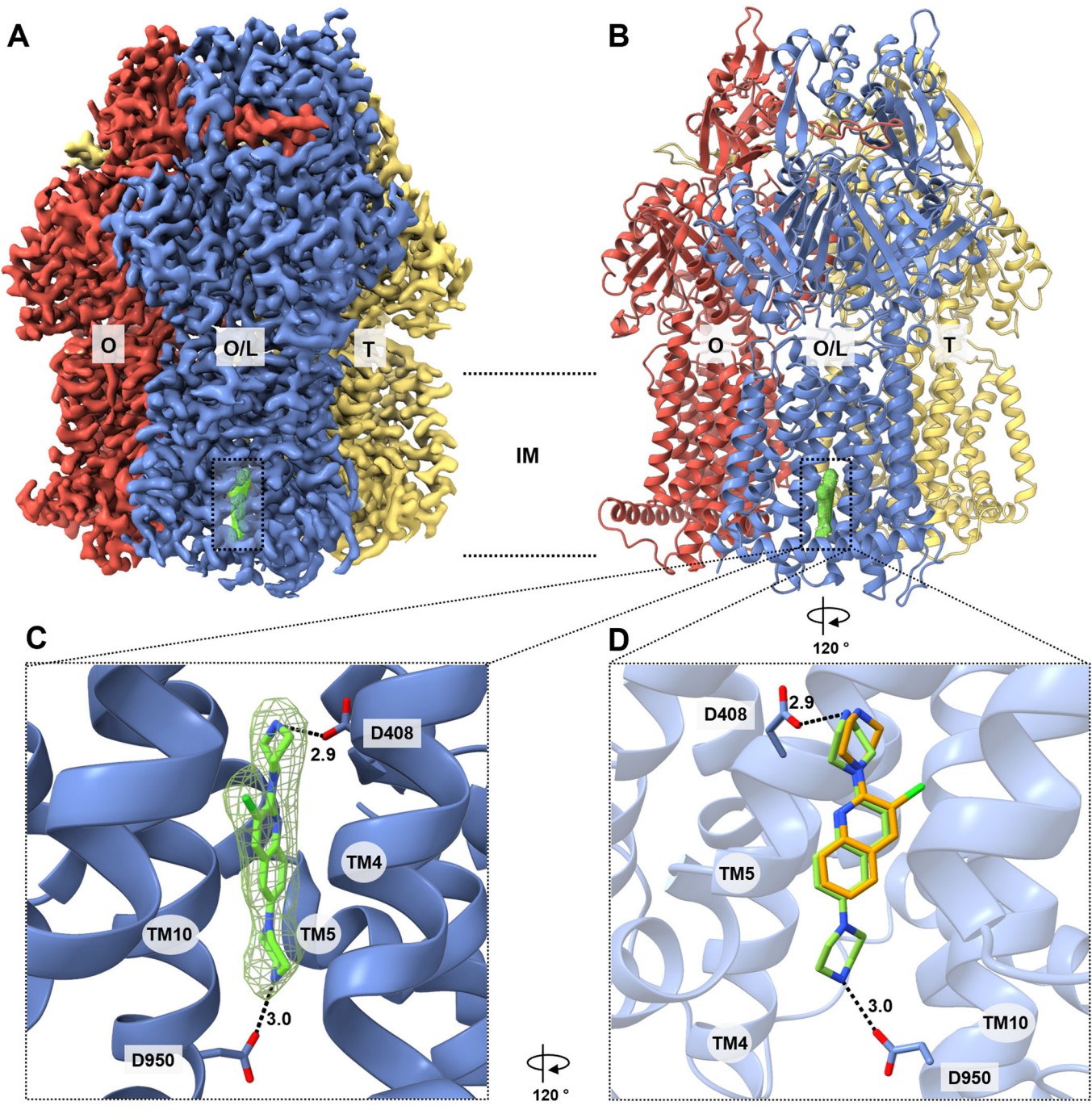

**Figure 1. Mechanism of action for BDM91288 (5).**

(A) Side view of the cryo-EM density map of *K. pneumoniae* AcrB in complex with BDM91288 (5) at an overall resolution of 2.97 Å. (B) Side view of the structure model of the AcrB trimer based on the electron densities in (A) in complex with BDM91288 (5). Densities and structures in **A** and **B** comprise the L (blue), T (yellow), and O (red) protomers and the BDM91288 (5) inhibitor binding to the TM domain of the L protomer is indicated in green colour. The L protomer in complex with the inhibitor adopts a transition state between the O state and L state. The boundaries of the inner membrane (IM) are indicated for the L protomer. (C) Enlarged view of BDM91288 (5) (green, stick representation, with green mesh for the observed density) binding site at the cytoplasmic side of the TM domain nested between transmembrane (TM) helices 4, 5, 10 of the AcrB L protomer (blue). (D) Superimposition of the binding position of BDM88855 (orange, stick representation) (Plé et al, 2022) and BDM91288 (5). The interacting residues (blue sticks, with oxygen atoms in red) forming salt bridges between the distal piperazine ring and D408, and the proximal piperazine ring with D950, are indicated by dashed lines and numbers represent the distance in Å. Further interactions include the halogen bond between the BDM91288 (5) chlorine and the main chain carbonyl oxygen of K939 (TM helix 10), as well as the hydrogen bond between the distal piperazine ring with the main chain carbonyl oxygen of L404 (TM helix 4) (Fig. EV2).

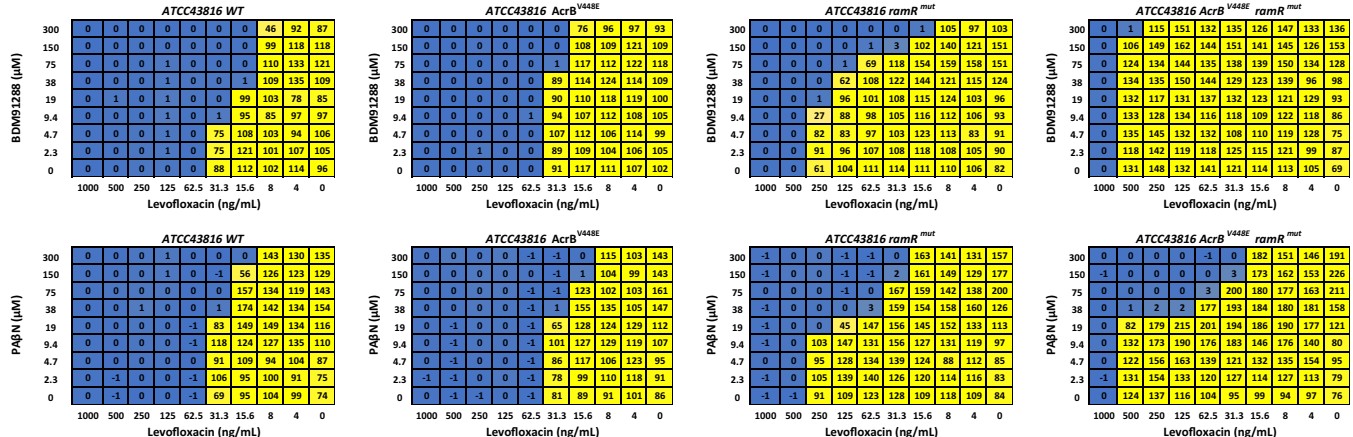

**Figure 2. Checkerboard assay evaluating levofloxacin antibiotic activity in the presence of different concentrations of BDM91288 (5) and PAβN.**

Susceptibility studies were performed on *K. pneumoniae* strain (WT), and isogenic mutants containing a point mutation in *acrB* (AcrB^V448E), *ramR* (RamR^mut = K124*stop) or both (AcrB^V448E RamR^mut). Values indicate the bacterial viability expressed as the median percentage of resazurine reduction (yellow cells >5%, blue cells <5%). Data represent the mean bacterial viability of at least three biological replicates. Source data are available online for this figure.

lysosomal trapping, which can play a major role for basic compounds (Aulin et al, 2022). In all compartments, BDM91288 (**5**) showed low clearance ($t_{1/2} > 10$ h), allowing for the effective maintenance of PyrPip concentrations over 24 h. Based on the PK data generated, it was predicted that BDM91288 (**5**) exposure in the lungs could be sufficient to demonstrate efficacy as a booster of antibiotic activity in a pulmonary model of *K. pneumoniae* infection.

### In vivo efficacy studies in *K. pneumoniae*-infected mice

As BDM91288 (**5**) presented a prolonged high PK exposure in mouse lungs, this compound was considered for in vivo proof-of-concept studies in a murine model of *K. pneumoniae* lung infection (Hoogerwerf et al, 2012). In this work, levofloxacin was chosen as an appropriate partner antibiotic because the ratio of the pharmacokinetic AUC to the MIC (i.e., AUC/MIC) best correlates with its in vivo efficacy (Scaglione et al, 2003). Therefore, by decreasing its MIC 4-fold on *K. pneumoniae* ATCC 43816 with 38 μM of BDM91288 (**5**) (Fig. 2) we expected an increase in the AUC/MIC ratio and a boost of levofloxacin efficacy when combined with this PyrPip in vivo. For this study, mice were infected intranasally with hypervirulent *K. pneumoniae* (ATCC 43816) and the bacterial loads were determined 4 h and 24 h post treatment by colony forming units in the lungs. Administration of a single dose of BDM91288 (**5**) (30 mg/kg, p.o.) showed no impact on the *K. pneumoniae* bacterial load in the lungs, while a single levofloxacin dose (10 or 50 mg/kg i.p) resulted in a 3.2- or 5.9-log reduction in lung bacterial load at 24 h compared to untreated mice (Fig. 3). Levofloxacin has a relatively short half-life in mice (Scaglione et al, 2003), which explains why for 10 mg/kg levofloxacin there is a rapid initial decrease in bacterial load, followed by a partial rebound when levofloxacin was eliminated from the body. To assess the in vivo efficacy of PyrPip, a single dose of BDM91288 (**5**) (30 mg/kg, p.o.) was combined with a single dose of levofloxacin (10 mg/kg, i.p.), and data showed that this combination resulted in a significantly improved bacterial clearance

compared to levofloxacin alone (Fig. 3). We also observed that the extent of levofloxacin boosting effect was inferior to that achieved with a 5-fold increase in levofloxacin dose (50 mg/kg). This could be due to a suboptimal PyrPip concentration in ELF that did not reach the effective dose required for a full boosting effect (Figs. 2 and EV3B). Together this data confirms the efficacy of BDM91288 (**5**) in mice and validates the potential of PyrPip-based efflux pump inhibitors for the boosting of antibiotic activity in vivo. Based on this proof-of-concept study, further development of PyrPips will be required to optimise inhibitor organ distribution as well as potency.

## Discussion

Tackling multidrug-resistant bacteria (MDR) has emerged as one of the greatest challenges of the 21st century, within which antibiotic-resistant *K. pneumoniae* strains pose a major clinical threat (Murray et al, 2022; Peirano et al, 2020). The development of novel antibiotics against Gram-negative infections usually fails due to the inability to achieve effective antibiotic concentrations inside the bacterial cell, which is highly dependent on the permeability of the outer membrane in combination with the activity of efflux pumps (Manrique et al, 2023).

The major clinically relevant efflux systems in Gram-negative bacteria belong to the RND superfamily (Schuldiner, 2018; Li et al, 2015). These RND-transporters, in addition to contributing to virulence and biofilm formation, efflux multiple classes of antibiotics out of the bacteria, playing a critical role in the development of antibiotic resistance (Li et al, 2015; Nishino et al, 2006; Alav et al, 2022). Overexpression of RND-efflux pumps through stress response mechanisms or accumulation of mutations in regulatory genes, has been shown to elevate antibiotic resistance, in some cases beyond clinical breakpoints. The relative importance of these regulators varies among species of Enterobacterales, with MarR/MarA and RamR/RamA being the major regulators of *acrAB* expression in *E. coli* and *K. pneumoniae*, respectively (Hentschke et al, 2010; Mao et al, 2020; Jiménez-Castellanos et al, 2016).

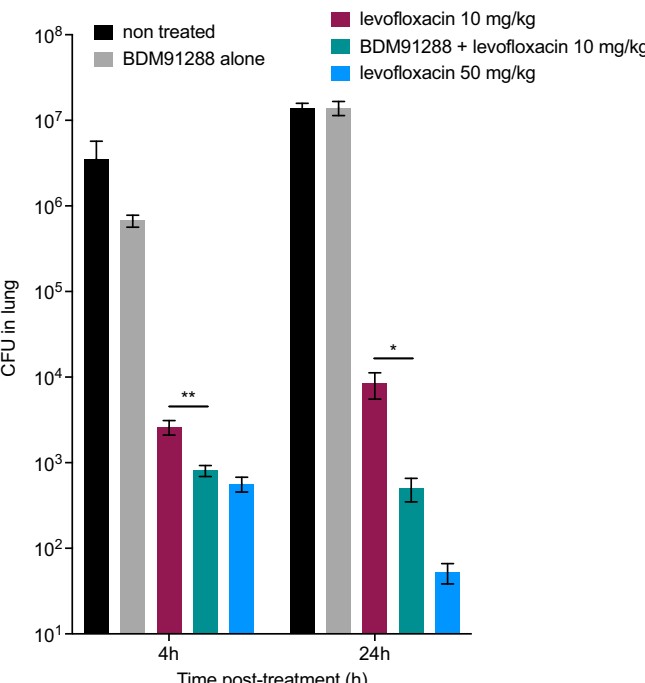

**Figure 3. BDM91288 (5) boosts the bactericidal effect of levofloxacin in a murine model of _K. pneumoniae_ lung infection.**

C57BL/6JRj mice ($n = 15$–24) were infected intranasally with $1 \times 10^4$ _K. pneumoniae_ ATCC 43816, treated or not 20 h post-infection with EPI BDM91288 (5) (30 mg/kg, orally) and 90 min later treated or not with levofloxacin (10 mg/kg or 50 mg/kg, intraperitoneally). Lungs of each animal were sampled at 4 h or 24 h post levofloxacin treatment to assess bacterial colonization. Plots show the mean ± SEM and represent a pool of 2 (24 h time point) or 3 (4 h time point) independent experiments. Statistical significance compared to levofloxacin 10 mg/kg group was assessed by the Mann–Whitney test (**$P = 0.0014$, *$P = 0.0333$). CFU: colony forming units. Source data are available online for this figure.

One strategy to fight antibiotic resistance caused by the expression of efflux pumps consists of developing EPIs to boost a wide range of antibiotics and restore drug susceptibility in multidrug-resistant Gram-negative clinical isolates, where resistance is associated to RND pump-mediated efflux. A number of AcrAB-TolC EPIs have been described in the literature (Compagne et al, 2023; Thakur et al, 2021; Moir et al, 2021; Lamut et al, 2019), though none have been approved for clinical use thus far. We have recently reported the discovery of a novel chemical family of PyrPip AcrB inhibitors which boost a large panel of AcrAB-TolC efflux pump substrates in _E. coli_ (Plé et al, 2022). PyrPips bind to a different location compared to previous EPIs (Compagne et al, 2023). While most of the EPIs bind to the antibiotic binding pockets in the periplasmic part of the pump, PyrPip inhibitors bind to the transmembrane domain of AcrB.

Here, we present a structure-based optimisation of our PyrPip series that led us to identify BDM91288 (5), a compound with suitable physico-chemical and pharmacokinetic properties, capable of potentiating the activity of a wide range of antibiotics in _K. pneumoniae_ and reverting antibiotic resistance in a _K. pneumoniae_ strain overexpressing _acrAB_ due to the absence of the RamR repressor. The co-structure of _K. pneumoniae_ AcrB with

BDM91288 (5) solved by cryo-EM revealed that this inhibitor forms, as expected, additional interactions with D950 in the TM domain with the $-NH_2^+-$ of the second piperazine moiety. Our combined efforts demonstrate that the rational extension of the BDM88855 (1) scaffold with additional moieties, including amino groups, can lead to enhanced PyrPip interactions within the transmembrane domain, thereby paving the way for the discovery of more potent compounds.

Multi-organ PK studies showed an accumulation of BDM91288 (5) in lung which guided us to prioritise the evaluation of its efficacy in a murine model of _K. pneumoniae_ lung infection. We then evaluated BDM91288 (5) in this infection model and showed that it was indeed able to potentiate the activity of levofloxacin in vivo. To our knowledge, this is the first time that an EPI has shown in vivo efficacy in a murine model of _K. pneumoniae_ lung infection. However, the combination of 10 mg/kg of levofloxacin with 30 mg/kg BDM91288 (5) was not as effective as 50 mg/kg levofloxacin suggesting incomplete inhibition of AcrB in this set up. This observation is in line with the measured PyrPip concentrations in the ELF which are below the concentrations required to fully boost levofloxacin activity in vitro. Therefore, future efforts will aim to further increase the potency of PyrPips and their organ distribution to overcome these challenges. In addition, the activity of levofloxacin was boosted a maximum of 4-fold in vitro, suggesting that the use of another antibiotic, better substrate of the AcrAB-TolC pump, could lead to greater efficacy in vivo in combination with a PyrPip compound. A comprehensive PK/PD analysis is thus needed to find a more suitable combination.

In conclusion, the combined medicinal chemistry, microbiology, and structural biology approach led to the new PyrPip-based allosteric efflux pump inhibitor BDM91288 (5), with enhanced pharmacokinetic properties while maintaining the multidrug boosting properties on Enterobacterales. Our in vivo proof-of-concept demonstrates unprecedented efficacy of this EPI in combination with levofloxacin on _K. pneumoniae_ infection and will guide our efforts to further optimise PyrPips potency and increase their concentration at the site of infection to maximise target engagement.

# Methods

## Medicinal chemistry

The chemical synthesis and structural validation of the compounds can be found in the Appendix.

## Bacterial strains and reagents

_E. coli_ BW25113 (https://ecoliwiki.org/colipedia/index.php/Category:Strain:BW25113) was obtained from Datsenko and Wanner on the authors request (Datsenko and Wanner, 2000). _E. coli_ C43(DE3) for overproduction of _Kp_AcrB was obtained from Miroux and Walker on the authors request (Miroux and Walker, 1996). The _acrB_ gene was amplified from _K. pneumoniae_ subsp. pneumoniae (FDAARGOS_775/ATCC 13883) genome, source: ATCC.

_E. coli_ BW25113, _K. pneumoniae_ ATCC 43816 (serotype 2) and its derivatives were cultured in LB (Difco), Cation Adjusted

Muller-Hinton (CAMHB, Difco), or Tryptic Soy broth (TS, Difco) at 37 °C. When necessary, appropriate concentration of given antibiotics were added to the cultures.

Antibiotics and PAβN were purchased from various vendors, including Sigma-Aldrich, Carbosynth Limited, Fisher Scientific, and Euromedex.

## Minimum inhibitory concentration (MIC) and checkerboard assay

MIC experiments were carried out according to the CLSI guidelines (CLSI 2023). Overnight cultures were diluted to an $OD_{600} = 0.001$ in CAMHB followed by addition of efflux pump inhibitors (EPIs) at a final concentration of 100 μM. Afterwards, 50 μL of bacterial suspension were transferred to 384-well flat-bottomed microtitre plates (Greiner) where a dose-response of the antibiotics of interest were added by serial dilutions (range = 0–128 mg/L) using a Tecan D300e Digital Dispenser (Tecan, France). Plates were incubated at 37 °C for up to 20 h. Bacterial viability was evaluated using the resazurin microtitre assay (REMA) and measured by fluorescence using a Tecan Spaert multimode plate reader (Ex: 530 nm Em: 590 nm). MICs were defined as the concentration that prevented 90% of resazurin turnover compared to the non-treated bacteria.

To evaluate the efficacy of BDM88855.HCl (1'), BDM91288 (5) and PAβN; their concentration-dependent boosting of levofloxacin activity was determined using a checkerboard assay. 384-well flat-bottomed microtitre plates containing a dose-response of each EPI (concentration range = 0–300 μM) and levofloxacin (concentration range = 0–1 mg/L) were prepared with a Tecan D300e Digital Dispenser (Tecan, France). Bacterial suspensions, incubation and plate readouts were carried out as described above.

## Genomic DNA extraction

Each bacterial culture was inoculated into a separate batch of 20 mL CAMHB in a stoppered flask of five times the culture volume to an initial $OD_{600} \approx 0.05$. Cultures were incubated with shaking (180 rpm) until the $OD_{600}$ had reached 0.5–0.7. Bacteria were collected by centrifugation at $4000 \times g$ and resuspended in 1 mL phosphate-buffered saline (PBS). Cells were transferred to a tube containing acid washed silica lysing matrix B (MP Biochemicals) and were disrupted in a FastPrep-24TM (MP Biomedicals). Lysates were collected and incubated with 4.5 μL of 20 mg/mL PureLink RNase A (ThermoFisher Scientific) for 10 min at room temperature. DNA was extracted as per standard phenol-chloroform-isoamyl alcohol (25:24:1) procedure (ThermoFisher Scientific). Genomic DNA and purity were checked by DeNovix DS-11 Series (DeNovix, Wilmington, Delaware), and Qubit4 (ThermoFisher, France).

## Whole-genome sequencing

Genomic DNA samples were sequenced by MicrobesNG (http://www.microbesng.com, Birmingham, UK) using an Illumina Novaseq600 platform with 150 bp paired end reads (mean coverage 30X). Samples were aligned to *K. pneumoniae* ATCC 43816 reference genome (Budnick et al, 2021). Genomic analysis was performed with Galaxy interface programme (https://usegalaxy.eu/) and allele variants were called with the software package Snippy v4.3.6 (https://github.com/tseemann/snippy).

## Isolation of BDM88855.HCl (1') resistant mutants

To select *K. pneumoniae* BDM88855.HCl (1') resistant clones, 50 μL of concentrated log-phase ATCC 43816 culture at $OD_{600} = 20$ were plated onto CAMHB agar containing either erythromycin (range = 1.25, 2.5, 5, 10, 20 or 40 mg/L) or the same erythromycin concentrations with 500 μM BDM88855.HCl (1'). Resistant colonies appeared within 24 h of incubation at 37 °C. For the first step of validation, several colonies were picked and grown in CAMHB without selective pressure, and antibiotic susceptibility determined (erythromycin, pyridomycin, levofloxacin, and linezolid) in the absence and presence of BDM88855.HCl (1') (100 μM) or BDM91288 (5) (100 μM).

## Construction of *K. pneumoniae ramR* and *acrB* point mutants

Mutations in *acrB* or *ramR* identified in the BDM88855.HCl (1') resistant mutant (a.k.a., KPJC11) were introduced individually into *K. pneumoniae* ATCC 43816 chromosome by allelic exchange (Cianfanelli et al, 2020). Briefly, 1-kb PCR products centred around each point mutation were generated by amplifying *acrB* with primers RH1251 (tatcgataagcttgatatcgGTGAACCAG-GACGGTTCC) and RH1252 (cggccgctctagaactagtgCTGCGA-GAAGTAGCCCATTG) or *ramR* with primers RH1249 (tatcgataagcttgatatcgACGCACTCATTATTAGGAAAGC) and RH1250 (cggccgctctagaactagtgCCAGCAGATCCTCGCTGA-TATC) from KPJC11 genomic DNA. PCR fragments were cloned into pFOK (Cianfanelli et al, 2020) using NEB-building kit (New England Biolabs) to generate pEP1533 and pEP1534. Correct plasmids were confirmed by sequencing and introduced into *K. pneumoniae* ATCC43816 WT via conjugation with the MFDpir + *E. coli* donor strain. Exconjugants were selected on kanamycin (50 mg/L) plates at 37 °C and plasmid chromosomal integration was checked by PCR. For each construction, a $Km^R$ clone was grown in 1 mL tryptone (1%), yeast extract (0.5%), sucrose (15%) broth for about 2 h at 37 °C followed by 1/1000 dilution into 1 mL of the previous broth but containing doxycycline (100 μg/L) and incubated overnight at 37 °C. The next day, cultures were diluted and plated onto LB agar. Isolated colonies were then patched onto LB plates with or without kanamycin (50 mg/L) and $Km^S$ candidates were evaluated for correct allelic exchange via *acrB* or *ramR* sequencing and then WGS to confirm no additional mutations were introduced.

## Berberine accumulation assay

The accumulation of the berberine AcrB substrate in *K. pneumoniae* WT and AcrB-V448E cells in the presence of different BDM91288 (5) concentrations was determined as described (Plé et al, 2022) with minor modifications. Briefly, an overnight LB pre-culture of each strain (0.6 mL) was used to inoculate a 30-mL LB culture and incubated for around 2 h (37 °C, 150 rpm) to reach OD600 around 1. Cells were harvested by centrifugation ($4000 \times g$, 10 min at 4 °C), washed three times with 10 mL of ice-cold buffer (50 mM potassium-phosphate pH 7.5 and 1 mM $MgSO_4$) and resuspended into the same buffer containing also 0.2% glucose to an $OD_{600}$ of 2. The bacterial suspension was then allowed to reach room temperature, after which 200 μL was

dispensed to wells of an assay ready 96-black-well plate (Greiner Bio-One) containing 2 µL of 100× concentrated serial dilution of BDM91288 (**5**). To initiate the uptake experiment, 2 µL of berberine 20 mM (final concentration of 200 µM) was added to each well and transferred to a microplate fluorescence plate reader (Spark, Tecan), where the kinetics of berberine uptake was monitored for 20 min at 30 °C (excitation 365 nm/emission 540 nm). Experiments were performed 4 independent times, and berberine uptake was calculated for all conditions as the rate of fluorescence increase over the first 10 min and expressed as RFU/min (mean ± SD). Fitting was done using GraphPad Prism 7.0 software.

## Cloning, expression, and purification of *K. pneumoniae* AcrB

*K. pneumoniae subsp. pneumoniae* (FDAARGOS_775/ATCC 13883) *acrB* gene was amplified using primers AcrB(NdeI)FW and AcrB(XhoI)RV and cloned into pET24a via NdeI/XhoI restriction/ligation cloning. Sequence verified pET24_KP_*acrB* was expressed in *E. coli* C43(DE3)*ΔacrAB* and purified similar to the protocol described previously (Plé et al, 2022). Briefly, 6 × 1 L LB medium supplemented with 1 mM MgSO$_4$ and 50 mg/L kanamycin was inoculated 1:400 with an *E. coli* C43(DE3)*ΔacrAB*/pET24_KP_*acrB* overnight culture grown from a single colony. Cultures were grown in 5 L shake flasks at 37 °C (135 rpm) until OD$_{600nm}$ reached 0.5–0.6, stored on ice for 30 min, induced with 0.5 mM (final) IPTG and incubated for further 20 h at 20 °C (135 rpm). All following purification steps were performed at 4 °C. Cells were harvested (6000 × *g*, 20 min) and resuspended in 3 volumes (w:v, final) lysis buffer (20 mM TRIS pH 8.0, 500 mM NaCl, 2 mM MgCl$_2$) supplemented with each 10 mg/L lysozyme and DNase for at least 30 min. The protease inhibitor PMSF (200 µM, final) was added directly before cell disruption with a pressure cell homogenizer (Stansted Fluid power Ltd., UK), 2x times at 1 bar pre-pressure. The lysate was cleared from debris at 20,000 × *g* for 20 min before subjected to preparative ultra-centrifugation at 142,000 × *g* for 60 min. Membranes were resuspended in resuspension buffer (20 mM TRIS pH 7.5, 300 mM NaCl, 10% glycerol) to a final concentration of 0.2 mg/mL, snap frozen in liquid nitrogen and stored at −80 °C. For protein purification 16 mL membrane suspensions (3.2 g) were thawed and solubilized at 4 °C in 1% DDM (w:v, 20% stock, (D-97002-C, Glycon)) and 12 mM imidazole (1 M stock, pH 7.5) in a final volume of 48 mL for at least 30 min under mild sample rotation. Buffer P1 (20 mM TRIS pH 7.5, 300 mM NaCl, 0.02% DDM) was used for volume adjustment. After ultra-centrifugation (161,000 × *g*, 30 min) the supernatant was rotated with 1 mL (2 mL 1:1 suspension) HisPur Ni-NTA resin (Thermo Scientific) for at least 30 min. The resin was washed in a gravity flow column with each 15 CV (15 mL) of P1 buffer supplemented with (i) 20 mM imidazole and (ii) 80 mM imidazole. The protein was eluted with 2 × 4 mL P1, 400 mM imidazole in a further volume of 8 mL P1. Eluates were concentrated in an Amicon Ultra-100 kDa cutoff spin concentrator to 700 µL and applied on a Cytiva Superose 6 increase 10/300 GL size exclusion column with 20 mM HEPES pH 7.5, 150 mM NaCl, 0.02% DDM at a flow rate of 0.25 mL/min. Trimeric protein fractions were collected (at an elution volume of approx. 13.5 mL) and concentrated in an Amicon Ultra-100 kDa cutoff spin concentrator to ~4.2 mg/mL.

### Primers

AcrB(NdeI)FW: atatatCATATGcctaatttctttatcgatcgc
AcrB(XhoI)RV: atatatCTCGAGatgatgctcaacctgatgg

## Cryo-EM sample preparation and data collection

For single-particle cryo-EM, 3.5 µL of purified *Kp*AcrB (4.2 mg/mL in 20 mM HEPES pH 7.5, 150 mM NaCl, 0.02% DDM) with added 165 µM of BDM91288 inhibitor without HCl (from 16 mM DMSO stock) were deposited onto glow-discharged Quantifoil R1.2/1.3, 300-mesh Cu holey carbon grids (Quantifoil Micro Tools GmbH) and vitrified in liquid ethane using a Vitrobot Mark IV (Thermo Scientific, Waltham, USA) at 100% relative humidity and 4 °C. Whatman blotting papers (grade 595) were pre-equilibrated in the Vitrobot chamber for 1 h at 100% relative humidity and 4 °C before plunge-freezing. A nominal blot force of −25, a wait time of 40 s, with blotting times of 8 s and 10 s were applied.

Dose-fractionated movies were acquired with EPU 3.1 (Thermo Scientific, Waltham, USA) at a nominal magnification of ×105,000 (0.837 Å per pixel) in nanoprobe EFTEM mode with a Titan Krios G3i cryo-TEM (Thermo Scientific, Waltham, USA) operating at 300 kV, equipped with a BioQuantum-K3 imaging filter (Gatan Inc., Pleasanton, USA), operated in zero loss peak mode with the energy slit width set to 30 eV. 8487 micrograph stacks with 50 frames per micrograph and a frame time of 0.05 s were recorded. The K3 camera was operated in counting mode with a dose rate of 19 (e-/Å$^2$ s$^{-1}$) and a total dose of 50 (e-/Å$^2$ s$^{-1}$) for a single micrograph was applied. The defocus range was set to −0.8 to −3.5 µm with a step size of 0.1 µm.

## Cryo-EM data processing

Cryo-EM data was processed with CryoSPARC v4.1.2 (Punjani et al, 2017). CryoSPARCS's own implementation was applied for beam-induced motion correction and CTF estimation. Initially, the blob picker using a particle diameter of 100–160 Å was applied for autopicking, 2D classes were generated and templates for automated template-picking were created. 1,396,333 particles were extracted after application of the template picker. False-positive picks and poor-quality particles were removed after two rounds of iterative unsupervised 2D classification. The remaining 380,181 particles were used as an input for an ab initio reconstruction with three classes. After ab initio reconstruction, 327,210 particles were selected, and local correction of beam-induced specimen movement was applied. The CTF was refined iteratively per group and non-uniform refinement with an applied C1-symmetry was used to refine the *Kp*AcrB homotrimer. Unsupervised 3D classification resulted in the further removal of poor-quality particles and the class (168,689 particles) showing the BDM91288 (**5**) binding site at the highest resolution of 3.01 Å was selected for further non-uniform refinement and local refinement resulting in a 2.97 Å density map.

### *Kp*AcrB model building and refinement

Swiss-Model (Waterhouse et al, 2018) was used for initial model generation. A structural model of *E. coli* AcrB (PDB ID: 7OUK) (Plé et al, 2022) was applied as a user template to create a starting

model based on the *K. pneumoniae* AcrB (FDAARGOS_775) amino acid sequence. In a Phenix real space refinement, the initial model was adapted to the densities of the cryo-EM map. The final structural model of *Kp*AcrB was iteratively built in Coot (Emsley and Cowtan, 2004) and alternating cycles of Phenix (Headd et al, 2012; Afonine et al, 2012) space refinement. The description for the assigned ligand (BDM91288 inhibitor) was taken from the Coot ligand builder. The identified lipid densities were representatively filled with phosphatidylglycerol 8:0/8:0 or phosphatidylglycerol 10:0/22:0, with the length of the alkyl chains chosen to match the assigned densities. Descriptions were taken from the Coot monomer dictionary and the Phenix elbow tool. The resulting structure was validated with Molprobity (Chen et al, 2010).

## Microsomal stability

To determine compound microsomal stability, liver microsomes from female mice (CD-1, Corning) were used. All incubations were performed in duplicate in a shaking water bath at 37 °C. The incubation mixtures contained 1 μM compound with 1% MeOH used as a vehicle, mouse liver microsomes (0.3 mg of microsomal protein per mL), 5 mM MgCl$_2$, 1 mM NADP, 5 mM glucose 6-phosphate, 0.4 U/mL glucose 6-phosphate dehydrogenase, and 50 mM potassium phosphate buffer (pH 7.4) in a final volume of 0.5 mL. Propranolol, known as a high hepatic clearance drug in rodents, was used as a quality-control compound. Controls (t$_0$ and t$_{final}$) were performed in the same conditions but without NADP. The reaction was initiated by the addition of the compound at 1 μM final. Aliquots were removed at 5, 10, 20, 30, and 40 min and the reaction was stopped by adding four volumes of ice-cold MeCN or MeOH containing an internal standard. The samples were centrifuged for 10 min at 12,000 rpm at 4 °C and the supernatants were transferred in matrix tubes for LC-MS/MS analysis. Each compound was quantified by converting the corresponding analyte/internal standard peak area ratios to percentage drug remaining, using the initial ratio values in control incubations as 100%.

Intrinsic clearance (CL$_{int}$) is expressed as μL/min/mg proteins. CL$_{int}$ = (dose/AUC$_\infty$)/[microsomes] where dose is the initial concentration of product in the incubate (1 μM), AUC$_\infty$ is the area under the concentration-time curve extrapolated to infinity and [microsomes] is the concentration in microsomes expressed in mg/μL.

## Solubility of BDM91288 (5)

0.5 mg of BDM91288 (**5**) were dissolved either in 1235 μL of PBS pH 7.4 (triplicate) or in 1235 μL of DMSO (triplicate twice). After gently stirring 24 h at room temperature, the solutions were centrifuged for 5 min at 4000 rpm and filtered over 0.45-μm filters (Millex-LH Millipore), excepted one of the DMSO triplicate. Then, 2 μL of each solution was diluted in 998 μL of MeOH were transferred in matrix tubes for LC-MS/MS analysis. The solubility was determined according to the following formula: Solubility (μM) = [AUC$_{(filtered\ PBS)}$/AUC$_{(not\ filtered\ DMSO)}$] × 1000.

## Plasma protein binding of BDM91288 (5)

In vitro plasma protein binding of BDM91288 (**5**) was measured in mouse plasma using equilibrium dialysis. The RED inserts were placed in a 48-well Teflon Plate (Pierce). Samples were prepared by mixing test compound with plasma at the appropriate concentration to reach a final drug concentration of 10 μM. Triplicate aliquots of plasma containing BDM91288 (**5**) at a concentration of 10 μM were pipetted to plasma side (red) of the insert and PBS pH 7.4 was placed into the receiver side (white) of the insert. The plate was covered with sealing tape and incubated in a 37 °C orbital shaker water bath for 4 h. Following incubation, samples were prepared in a mixed matrix configuration. Aliquots of samples were pipetted into 96-well plates and precipitation buffer was added to precipitate the proteins. Samples were mixed by vortexing, then centrifuged for 10 min at 12,000 rpm at 4 °C. The supernatant was assayed directly by LC-MS/MS. The following equation was used to calculate the percentage of bound drug fraction (% PPB) using this equilibrium dialysis method: % unbound = ([Buffer chamber]/[Plasma chamber] × 100.

## hERG inhibition of BDM91288 (5)

hERG (human ether-a-go-go-related gene) inhibition was measured at Eurofins Cerep France using hERG Human Potassium Ion Channel [3H] Dofetilide Binding (Antagonist Radioligand) Assay.

## Cytotoxicity of BDM91288 (5)

BALB/3T3 cells (CCL-163 from ATCC) were not authenticated. The cell lines were tested negative for mycoplasma contamination (kit mycoalert LONZA). The cytotoxicity of compound BDM91288 (**5**) was determined on BALB/3T3 cells using live imaging following both Hoechst 33342 and propidium iodide staining. Propidium iodide is a not membrane-permeable DNA stain, allowing to differentiate necroptic, apoptotic and healthy cells based on loss of membrane integrity. Briefly, BALB/3T3 cells were seeded in 384-well plate (3000 cells per well in 40 μL DMEM containing 10% foetal calf serum), and 24 h later, compound BDM91288 (**5**) was added (0, 12.5, 25, 50, and 100 μM) to the culture medium, as well as Hoechst 33342 (20 ng/mL) and propidium iodide (1 μg/mL). Final medium volume was 70 μL. 24 h after compounds addition, live imaging was performed using the automated confocal microscope In Cell Analyzer 6000 (GE Healthcare). The cytotoxicity was defined as the percent of cells labelled by propidium iodide in the total cell population (Hoechst labelled cells) using Columbus software (Perkin Elmer). Compounds were tested in triplicate. Carfilzomib (499, 249, 142, 61, and 30.6 nM) was used as a positive control in this assay.

## Pharmacokinetic of BDM91288 (5) in mice

### Animals
Six-week-old female CD-1 mice were purchased from Charles River (Saint Germain Nuelles, France). Two different mice strains were used for PK (CD-1) and efficacy (C57BL/6JRj) studies as it was shown in the literature that there was little difference in the PK parameters measured between mice strains (Barr et al, 2020). All animals were maintained in standard animal cages under conventional laboratory conditions (12 h/12 h light/dark cycle, 22 °C) with ad libitum access to food and water. The animals were maintained in compliance with European standards for the care and use of laboratory animals and experimental protocols approved by the

local Animal Ethical Committee (agreement no. APAFIS#27003-2020082815505003).

## Pharmacokinetics

### Animal dosing and tissue sampling

BDM91288 (**5**) was dissolved in 10% kleptose hydroxypropyl-β-cyclodextrin in distilled water and administered per os at 30 mg/kg to female CD-1 mice (~25–30 g). The concentration of BDM91288 (**5**) in plasma and lungs was measured at different time points after administration of a single dose of compounds. Three mice per time point were anesthetized with isoflurane and blood taken from the retro-orbital sinus using sampling heparinated tubes (4 °C). The blood samples were centrifuged (3500 rpm, 20 min, 4 °C) for plasma separation and stored at −80 °C before BDM91288 measurement. After rodent sacrifice by cervical dislocation, lung was washed in sodium chloride solution (0.9%), frozen in liquid nitrogen and stored at −80 °C.

In a separate experiment, mice were simultaneously administered with BDM91288 (**5**) (p.o., 30 mg/kg in 10% kleptose hydroxypropyl-β-cyclodextrin) and with levofloxacin (i.p., 10 mg/kg in water for injection). The concentration of compounds in plasma, lungs and ELF was measured after 2 h, 6 h and 24 h. Four mice per time point were anesthetized with isoflurane and blood taken from the retro-orbital sinus using sampling heparinated tubes (4 °C). The blood samples were centrifuged (3500 rpm, 20 min, 4 °C) for plasma separation and stored at −80 °C before compound measurement. After rodent sacrifice by cervical dislocation, bronchoalveolar lavage (BAL) fluid was collected using a cannula inserted into the trachea. 1 mL of PBS was introduced in lungs via the cannula and the collected BAL fluid was centrifuged (5 min, 3500 rpm, 4 °C). Supernatants were removed, their volume recorded, and samples stored at −80 °C until analysis. Remaining lung was washed in sodium chloride solution (0.9%), frozen in liquid nitrogen and stored at −80 °C.

### Compound concentration determination in plasma, lung, BAL

Plasma and BAL fluid samples were thawed on ice. Aliquots were precipitated with ice-cold MeCN/MeOH 50:50 (1 to 10 ratio) containing propranolol (100 nM) as internal standard. The samples were vigorously mixed with a vortex and centrifuged for 10 min at 12,000 rpm, 4 °C, and the supernatants were transferred into Matrix tubes for LC-MS/MS analysis. Lungs were homogenized in ultrapure water, in a 1 to 4 ratio (W/V) (using a Tissue Lyzer II from Qiagen); then compounds were extracted with a MeOH/MeCN 50:50 mixture in a 1 to 1 ratio (V/V). After centrifugation (12,000 rpm, 10 min, 4 °C) of the homogenate samples, supernatants were diluted (1 to 10) with a MeOH/MeCN 50:50 mixture containing propranolol (100 nM) as internal standard and transferred into Matrix tubes for LC-MS/MS analysis. Spiked standard solutions were prepared the same way in control matrix.

Samples were analysed using a UPLC system Acquity I Class (Waters), combined with a triple quadrupole mass spectrometer Xevo TQD (Waters). The column, placed at 40 °C, was an Acquity BEH C18 50 × 2.1 mm, 1.7 μm column (Waters) and the following mobile phases were used: $H_2O$ 0.1% HCOOH as solvent A and MeCN 0.1% HCOOH as solvent B at a flow rate of 0.6 mL/min. The gradient was initiated at 2% B, maintained for 10 s then increased linearly to 98% B in 1′50 and maintained at 98% B for 30 s before returning to initial conditions. The mass spectrometer was

equipped with an electrospray ionisation source with the following parameters: polarity ES+, capillary 0.5 kV, desolvation temperature 600 °C, source temperature 150 °C, cone gas flow 50 L/h, desolvation gaz flow 1200 L/h; the compounds were detected in MRM mode with the following transitions: BDM91288 332.1-70.0 (cone voltage: 48 V; collision energy: 32 eV), levofloxacin 362.1-261.1 (cone voltage: 40 V; collision energy: 28 eV). Data analysis and processing were performed using MassLynx software (Waters).

### Compound concentration determination in ELF

The concentration of compound in ELF, [cpd]ELF, was calculated as follows:

$$[cpd]_{ELF} = [cpd]_{BAL} \times V_{BAL}/V_{ELF}$$

where $[cpd]_{BAL}$ is the measured concentration of compound in BAL fluid supernatant, $V_{BAL}$ and $V_{ELF}$, the volume of aspired BAL and the calculated volume of ELF, respectively.

The volume of ELF was calculated as follows:

$$V_{ELF} = V_{BAL} \times [Urea]_{BAL}/[Urea]_{Plasma}$$

where $V_{ELF}$ represents the apparent volume of ELF, $V_{BAL}$ is the volume of aspired BAL, and $[Urea]_{BAL}$ and $[Urea]_{Plasma}$ are concentration of urea in BAL fluid or plasma, respectively.

Concentrations of urea in plasma and in BAL fluid supernatant was measured using the UREA Nitrogen (BUN) colorimetric detection kit, according to the supplier procedural guidelines (Invitrogen).

### *K. pneumoniae* mouse lung infection

Mice experiments complied with national, institutional and European regulations and ethical guidelines, were in accordance with ARRIVE guidelines 2.0 and were approved by the Institutional Animal Care and Use Committee (animal facility agreement D59-350009, Institut Pasteur de Lille, protocol reference: APA-FIS#25953, 2020060916081461_v2). Six to eight weeks old female C57BL/6JRj mice were purchased from Janvier Laboratories (Saint Berthevin, France), allocated into experimental groups in a random fashion and maintained in individually ventilated cages (Innorack® IVC Mouse 3.5) and handled in a vertical laminar flow biosafety cabinet (Class II Biohazard, Tecniplast). All infections were performed in an animal biosafety level 2 facility.

Inoculum was prepared as follows. Tryptic Soy broth was inoculated with *K. pneumoniae* from a frozen stock and grown overnight at 37 °C without shaking. 1 mL of this overnight bacterial culture was transferred to a warm fresh Tryptic Soy broth and grown at 37 °C until it reached the mid-log. Cells were pelleted, washed with PBS and resuspended to the appropriate concentration.

Infections were performed by intranasal (i.n.) route in mice that were previously anesthetized by intraperitoneal (i.p.) injection of 1.25 mg of ketamine and 0.25 mg of xylazine in 200 μL of PBS. Mice (22 for none treated at 4 h, 22 for BDM91288 (**5**) alone at 4 h, 24 for levofloxacin 10 mg/kg at 4 h, 24 for levofloxacin + BDM91288 (**5**) at 4 h, 22 for levofloxacin 50 mg/kg at 4 h, 15 for none treated at

 

## The paper explained

### Problem

Antimicrobial resistance has become a major challenge for global health, jeopardising our ability to treat infectious diseases effectively. One of the factors contributing to the increase in antimicrobial resistance is the overuse and misuse of antibiotics in human medicine, agriculture, and veterinary practices that led to the selection of bacteria resistant to a plethora of antibiotics. The consequences of antimicrobial resistance are considerable, resulting in increased morbidity and mortality rates, prolonged illness duration, and higher healthcare costs. This threat has led the World Health Organisation to prioritise research and development of antibiotics for drug-resistant Enterobacterales such as *Escherichia coli* and *Klebsiella pneumoniae*. A major barrier for developing novel antibiotic classes for Gram-negative bacteria is getting sufficient compound into these bacteria to kill them, as very efficient broad-specificity efflux pumps can extrude various antibiotics from the bacteria, which results in the persistence of the bacterial infection.

### Results

In this work, we developed a novel class of inhibitors, acting on the *K. pneumoniae* AcrAB-TolC efflux pump, to increase the concentration of antibiotics inside the bacterium and thus potentiate their activity. We optimised the physico-chemical and pharmacokinetic properties of these efflux pump inhibitors to increase their availability at the site of infection. In this way, we identified a compound, BDM91288, with such favourable drug-like properties for in vivo efficacy studies. This compound was able to potentiate the activity of a panel of antibiotics against *K. pneumoniae* as well as revert clinically relevant antibiotic resistance mediated by efflux pump overexpression. The target of this inhibitor was confirmed to be the efflux pump AcrB through the isolation and generation of resistant *K. pneumoniae* mutants. The binding mode of BDM91288 to an allosteric pocket of *K. pneumoniae* AcrB was validated by structural biology analysis using single-particle cryo-EM. Finally, proof-of-concept studies demonstrated that oral administration of BDM91288 significantly potentiated the in vivo efficacy of the antibiotic levofloxacin in a murine model of *K. pneumoniae* lung infection.

### Impact

With the increasing global threat of antibiotic resistance of Gram-negative bacteria, a promising strategy for combating efflux-mediated resistance is to combine existing antibiotics with adjuvant antimicrobial agents such as efflux pump inhibitors. Overexpression of antibiotic efflux pumps can lead to clinically relevant broad-spectrum antibiotic resistance, and the development of an efflux pump inhibitor that disrupts such efflux will revert susceptibility to many currently used antibiotics and effectively reduce the burden of antimicrobial resistance. As a consequence, this will result in reducing fatality rate, an improved patient treatment and a considerable reduction in healthcare costs. Since the inactivation of the broad-spectrum AcrB efflux pump will affect the efficacy of multiple antibiotics, this new therapeutic approach will improve a great many existing antibiotic treatments and extend their lifespan. Proof-of-concept studies in mice with the efflux pump inhibitor BDM91288 pave the way for the discovery of more potent allosteric efflux pump inhibitors, that could be used as adjuvants in antibiotic treatment.

24 h, 16 for BDM91288 (**5**) alone at 24 h, 16 for levofloxacin 10 mg/kg at 24 h, 16 for levofloxacin + BDM91288 (**5**) at 24 h, 16 for levofloxacin 50 mg/kg at 4 h) were infected i.n. with *K. pneumoniae* using $4 \times 10^4$ CFU of a fresh inoculum diluted in 30 μL of PBS. 22 h post-infection, mice were treated with BDM91288 (**5**) (30 mg/kg, 200 μL, formulated in 10% hydroxypropyl-beta-cyclodextrine (Kleptose HP; Roquette) at 3 mg/mL) by intragastric route. Levofloxacin (10 mg/kg or 50 mg/kg in 200 μL of water for injection) was administrated intraperitoneally (i.p.) 90 min later.

At 4 h or 24 h post antibiotic treatment, mice were euthanized by cervical dislocation. Lungs were collected in 1 mL of PBS, homogenized with a T-25 digital Ultraturrax® (IKA). CFU counts were determined by plating serial dilutions onto Tryptic Soy agar plate.

## Statistical analysis

Results are expressed as mean as indicated in figure legends. Intergroup differences were analysed using the Mann–Whitney test with GraphPad Prism 9.0 software. Results were considered significant for $P < 0.05$ indicated by "*" and $P < 0.01$ indicated by "**". Exclusion criteria include mice without signs of infection or mice that died prior to analysis. No blinding was performed.

## Data availability

The cryo-electron microscopy density map of *Kp*AcrB (2.97 Å) has been deposited in the EM Data Base under the accession number EMD-17350 (https://www.ebi.ac.uk/emdb/EMD-17350). Atomic coordinates have been deposited in the Protein Data Bank under the accession code 8P1I. Whole-genome sequencing data (fastq files) for parental and mutant *K. pneumoniae* strains studies in this work have been deposited at NCBI (BioProjectID: PRJNA1017663). Materials are available on request from the corresponding authors under a material transfer agreement. The synopsis graphic was created with BioRender.com.

## Peer review information

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

## Acknowledgements

We thank Tom van der Poll (UMC, Amsterdam, The Netherlands) for advice on mouse model of *K. pneumoniae* lung infection and the gift of *K. pneumoniae* ATCC 43816 and Valérie Landry for cytotoxicity experiments. This research was financially supported by l'Agence Nationale de la Recherche (ANR, France) in partnership with the Bundesministerium für Bildung und Forschung (BMBF, Germany) (program EFFORT, ANR-19-AMRB-0007 (RCH, MF), BMBF-16GW0236K/01KI2123 (KMP, ASF). Research was further supported by Feder (12001407 (D-AL) Equipex Imaginex BioMed), ATIP-Avenir, Institut National de la Santé et de la Recherche Médicale, Centre National de la Recherche Scientifique, Université de Lille, Institut Pasteur de Lille, Région Hauts-de-France. Virginie Meurillon received a doctoral grant from the Fondation pour la Recherche Médicale (FRM: ECO202206015500). The NMR facilities were funded by the Région Hauts-de-France (France), the Ministère de la Jeunesse, de l'Education Nationale et de la Recherche (MJENR), the Fonds Européens de Développement Régional (FEDER) and the Université de Lille. We thank Roquette Frères for the gift of Kleptose HP as well as ARIADNE-criblage (UMS2014-UAR2014-PLBS) and ARIADNE-ADME (French national infrastructure ChemBioFrance) for providing access to their facilities and procedures. We thank the Central Electron Microscopy Facility at the MPI of Biophysics in Frankfurt, which enabled us to collect the *Kp*AcrB dataset, particularly Sonja Welsch who assisted during the data collection.

## Author contributions

**Anais Vieira Da Cruz**: Conceptualization; Data curation; Formal analysis; Validation; Investigation; Visualization; Methodology; Writing—original draft. **Juan-Carlos Jiménez-Castellanos**: Conceptualization; Data curation; Formal analysis; Validation; Investigation; Visualization; Methodology; Writing—original draft; Writing—review and editing. **Clara Börnsen**: Conceptualization; Data curation; Formal analysis; Validation; Investigation; Visualization; Methodology; Writing—original draft; Writing—review and editing. **Laurye Van Maele**: Conceptualization; Resources; Data curation; Formal analysis; Validation; Investigation; Visualization; Methodology; Writing—original draft. **Nina Compagne**: Conceptualization; Data curation; Formal analysis; Investigation; Methodology; Writing—review and editing. **Elizabeth Pradel**: Conceptualization; Data curation; Formal analysis; Validation; Investigation; Methodology; Writing—review and editing. **Reinke T Müller**: Conceptualization; Data curation; Formal analysis; Investigation; Methodology. **Virginie Meurillon**: Conceptualization; Data curation; Formal analysis; Investigation; Methodology. **Daphnée Soulard**: Formal analysis; Investigation; Methodology. **Catherine Piveteau**: Resources; Data curation; Formal analysis; Validation; Investigation; Methodology; Writing—original draft. **Alexandre Biela**: Resources; Data curation; Formal analysis; Investigation; Methodology. **Julie Dumont**: Resources; Data curation; Formal analysis; Investigation; Methodology. **Florence Leroux**: Resources; Data curation; Formal analysis; Validation; Investigation; Methodology; Writing—original draft. **Benoit Deprez**:

Conceptualization; Writing—review and editing. **Nicolas Willand**: Conceptualization; Writing—review and editing. **Klaas M Pos**: Conceptualization; Data curation; Formal analysis; Supervision; Funding acquisition; Validation; Visualization; Writing—original draft; Project administration; Writing—review and editing. **Achilleas S Frangakis**: Conceptualization; Data curation; Supervision; Funding acquisition; Validation; Visualization; Writing—original draft; Project administration; Writing—review and editing. **Ruben C Hartkoorn**: Conceptualization; Data curation; Formal analysis; Supervision; Funding acquisition; Validation; Visualization; Writing—original draft; Project administration; Writing—review and editing. **Marion Flipo**: Conceptualization; Data curation; Formal analysis; Supervision; Funding acquisition; Validation; Visualization; Writing—original draft; Project administration; Writing—review and editing.

## Disclosure and competing interests statement

AVDC, J-CJ-C, NC, RTM, KMP, MF, NW and RCH are inventors on patent application covering the EPI described in this manuscript. The remaining authors declare no competing interests.

# Expanded View Figures

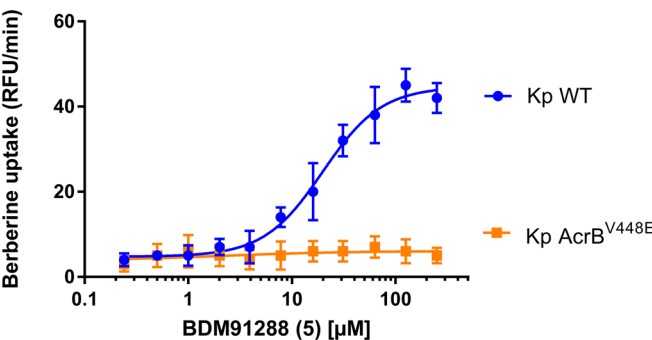

**Figure EV1. Rate of berberine uptake by *K. pneumoniae* WT and *K. pneumoniae* AcrB[V448E] in dependence of increasing BDM91288 (5) concentrations.**

Data are a mean and SD of four independent experiments. Source data are available online for this figure.

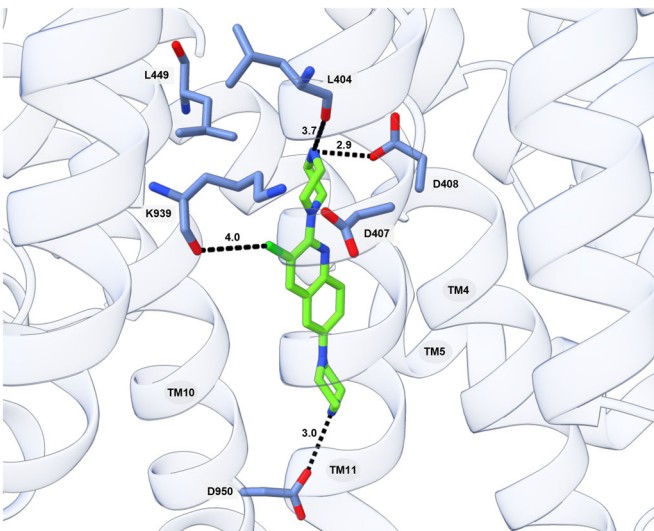

**Figure EV2.   Structure of *Kp*AcrB in complex with BDM91288 (5).**

Enlarged view of the inhibitor binding site showing interacting residues (blue sticks) in less or equal than 4 Å distance from BDM91288 (**5**). The salt bridge between the distal piperazine ring and D408 (TM helix 4), the halogen bond between the BDM91288 (**5**) chlorine and the K939 (TM helix 10) main chain carbonyl oxygen, the hydrogen bond between the L404 (TM helix 4) main chain carbonyl oxygen and the distal piperazine ring, and the salt bridge between D950 (TM helix 10) and the proximal piperazine ring are indicated by dashed lines and numbers represent the distance in Å.

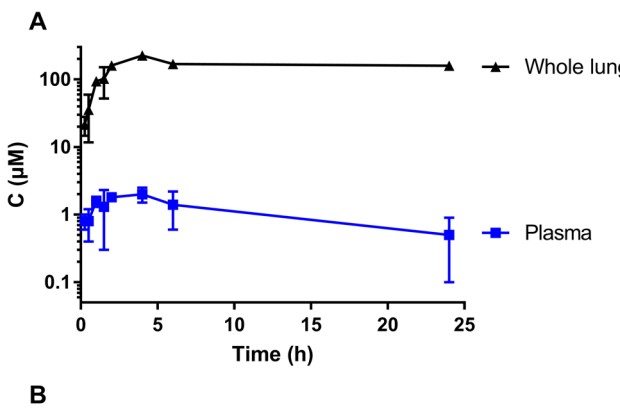

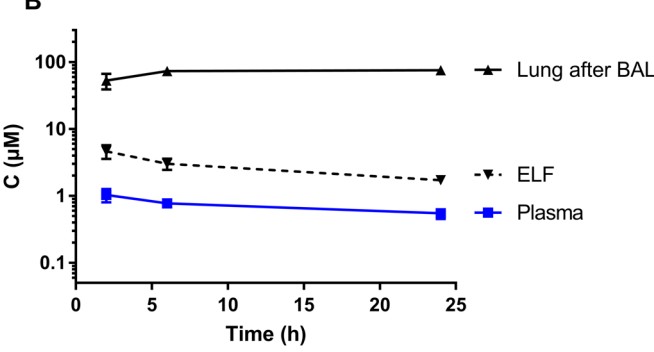

**Figure EV3. Pharmacokinetic profile of a single dose BDM91288 (5) (30 mg/kg, orally, formulated in 10% hydroxypropyl-β-cyclodextrin) in mice.**

(A) Concentration of BDM91288 (5) versus time in whole lung and plasma. Plots show the mean ± SD ($n = 3$). (B) Concentration of BDM91288 (5) versus time in lung after bronchoalveolar lavage (BAL), epithelial lining fluid (ELF) and plasma. BDM91288 (5) was given in combination with a single dose levofloxacin (10 mg/kg in water, i.p.). Plots show the mean ± SD ($n = 4$). Source data are available online for this figure.

