## [Peer Review File · EMBO Molecular Medicine]

Pyridylpiperazine efflux pump inhibitor boosts in vivo antibiotic efficacy against *K. pneumoniae*

Anais Vieira Da Cruz, Juan-Carlos Jiménez-Castellanos, Clara Börnsen, Laurye Van Maele, Nina Compagne, Elizabeth Pradel, Reinke Müller, Virginie Meurillon, Daphnée Soulard, Catherine Piveteau, Alexandre Biela, Julie Dumont, Florence Leroux, Benoit Deprez, Nicolas Willand, Klaas Pos, Achilleas Frangakis, Ruben Hartkoorn, and Marion Flipo

DOI: [10.15252/emmm.202318236](https://doi.org/10.15252/emmm.202318236)

Corresponding authors: Marion Flipo (marion.flipo@univ-lille.fr) , Achilleas Frangakis (frangak@biophysik.org), Klaas Pos (pos@em.uni-frankfurt.de)

Review Timeline:

Submission Date:	3rd Jul 23
Editorial Decision:	14th Aug 23
Revision Received:	22nd Sep 23
Editorial Decision:	2nd Nov 23
Revision Received:	9th Nov 23
Accepted:	17th Nov 23

Editor: Poonam Bheda

Transaction Report:

14th Aug 2023

Dear Dr. Flipo,

Thank you for the submission of your manuscript to EMBO Molecular Medicine. We have now received feedback from the three reviewers who agreed to evaluate your manuscript. As you will see from the reports below, the referees acknowledge the interest of the study and are overall supporting publication of your work pending appropriate revisions.

Addressing the reviewers' concerns in full will be necessary for further considering the manuscript in our journal, and acceptance of the manuscript will entail a second round of review. EMBO Molecular Medicine encourages a single round of revision only and therefore, acceptance or rejection of the manuscript will depend on the completeness of your responses included in the next, final version of the manuscript. For this reason, and to save you from any frustrations in the end, I would strongly advise against returning an incomplete revision.

We are expecting your revised manuscript within three months, if you anticipate any delay, please contact us.

We require:

4) A .docx formatted letter INCLUDING the reviewers' reports and your detailed point-by-point responses to their comments. As part of the EMBO Press transparent editorial process, the point-by-point response is part of the Review Process File (RPF), which will be published alongside your paper.

5) A complete author checklist, which you can download from our author guidelines (<https://www.embopress.org/page/journal/17574684/authorguide#submissionofrevisions>). Please insert information in the checklist that is also reflected in the manuscript. The completed author checklist will also be part of the RPF.

6) Please note that all corresponding authors are required to supply an ORCID ID for their name upon submission of a revised manuscript.

7) It is mandatory to include a 'Data Availability' section after the Materials and Methods. Before submitting your revision, primary datasets produced in this study need to be deposited in an appropriate public database, and the accession numbers and database listed under 'Data Availability'. Please remember to provide a reviewer password if the datasets are not yet public (see <https://www.embopress.org/page/journal/17574684/authorguide#dataavailability>).

In case you have no data that requires deposition in a public database, please state so in this section. Note that the Data Availability Section is restricted to new primary data that are part of this study. This study includes no data deposited in external repositories.

8) For data quantification: please specify the name of the statistical test used to generate error bars and P values, the number (n) of independent experiments (specify technical or biological replicates) underlying each data point and the test used to calculate p-values in each figure legend. The figure legends should contain a basic description of n, P and the test applied. Graphs must include a description of the bars and the error bars (s.d., s.e.m.). Please provide exact p values.

9) Our journal encourages inclusion of *data citations in the reference list* to directly cite datasets that were re-used and obtained from public databases. Data citations in the article text are distinct from normal bibliographical citations and should directly link to the database records from which the data can be accessed. In the main text, data citations are formatted as

follows: "Data ref: Smith et al, 2001" or "Data ref: NCBI Sequence Read Archive PRJNA342805, 2017". In the Reference list, data citations must be labeled with "[DATASET]". A data reference must provide the database name, accession number/identifiers and a resolvable link to the landing page from which the data can be accessed at the end of the reference. Further instructions are available at .

13) Author contributions: CRediT has replaced the traditional author contributions section because it offers a systematic machine readable author contributions format that allows for more effective research assessment. Please remove the Authors Contributions from the manuscript and use the free text boxes beneath each contributing author's name in our system to add specific details on the author's contribution. More information is available in our guide to authors.

Please also suggest a striking image or visual abstract to illustrate your article as a PNG file 550 px wide x 300-600 px high. Share synopsis text and image, as well as eTOC:

Please note that these would be the final versions and changes during proofing are usually not allowed

16) As part of the EMBO Publications transparent editorial process initiative (see our Editorial at <http://embomolmed.embopress.org/content/2/9/329>), EMBO Molecular Medicine will publish online a Review Process File (RPF) to accompany accepted manuscripts.

In the event of acceptance, this file will be published in conjunction with your paper and will include the anonymous referee reports, your point-by-point response and all pertinent correspondence relating to the manuscript. Let us know whether you agree with the publication of the RPF and as here, if you want to remove or not any figures from it prior to publication.

I look forward to receiving your revised manuscript.

Yours sincerely,

Poonam Bheda

Poonam Bheda, PhD
Scientific Editor
EMBO Molecular Medicine

Please note: When submitting your revision you will be prompted to enter your funding and payment information. This will allow Wiley to send you a quote for the article processing charge (APC) in case of acceptance. This quote takes into account any reduction or fee waivers that you may be eligible for. Authors do not need to pay any fees before their manuscript is accepted and transferred to the publisher.

EMBO Press participates in many Publish and Read agreements that allow authors to publish Open Access with reduced/no publication charges. Check your eligibility: <https://authorservices.wiley.com/author-resources/Journal-Authors/open-access/affiliation-policies-payments/index.html>

***** Reviewer's comments *****

Referee #1 (Remarks for Author):

This study is focused on a preliminary characterization of an efflux pump inhibitor of a pyridylpiperazine class of molecules. These inhibitors were previously reported. Here, the authors carried out focused medicinal chemistry efforts to improve the metabolic stability of the previously reported hit, to evaluate its resistance potential and limited PK and in vivo studies. The compound 5 showed improved stability and an accumulation in lungs, but no improvement in the efflux inhibitory activity. The resistance markers were linked to a point mutation in the target AcrB and its regulatory system RamRA. The cryo-EM structure of the target AcrB from *K. pneumoniae* bound to 5 further validated the result. The compound potentiated a fluoroquinolone antibiotic levofloxacin in the *K. pneumoniae* pulmonary infection model.

Overall, studies are well-designed and carried out, the results are convincing and demonstrate that this class of compounds has a desired effect on target. On the other hand, the study has also showed that compounds have a resistance liability that will limit their further development for clinical use. Another weakness is that the demonstrated activity is in a combination with levofloxacin. However, in clinics, the primary reason for fluoroquinolone resistance are mutations in topoisomerases, the target of these antibiotics. The reported compounds will have no clinical utility if further developed in a combination with levofloxacin.

Referee #2 (Remarks for Author):

The manuscript "Pyridylpiperazine efflux pump inhibitor boosts in vivo antibiotic efficacy against *K. pneumoniae*" by Cruz and coworkers describes the medicinal chemistry optimization and characterization of a bacterial efflux pump inhibitor (EPI). The work covers both the synthesis of a range of pyridylpiperazine derivatives and they are tested against a variety of *K. Pneumonia* strains that have increased AcrAB/TolC expression in the presence of a range of clinically used antibiotics. Their lead compound in this study (BDM91288) was then interrogated by single particle cryoEM against the AcrB trimer and its structure activity relationship was determined. Furthermore, this compound was then tested in a murine model lung infection study where in combination with levofloxacin, it showed better bacterial clearance than the antibiotic alone.

In my opinion this manuscript is of excellent quality and is a well-conceived and well carried out study that is of great import in the further development of new antibacterial treatments. However, I do have some minor comments, mostly based on the cryoEM analysis. I also want to thank the authors in sharing their data.

1. There are some minor modelling problems, one example is in ChainA; Y467,F459,L875 are all in wrong density, which has led to some Ramachandran outliers. This is true across the whole structure, so it would be ideal to carefully go through and ensure no other errors.
2. In the published structure 7OUK had piperazine compound in two positions. I note that there is weak density in Chain C (W808) (in the authors structure) for second allosteric site. Perhaps some focussed refinements on this area might also elucidate this second site.
3. While I completely agree with the authors placement of BDM91288 in ChainA, there are lots of other unaccounted for density (most of which appears to be various lipids, or detergent molecules). However, there are some other densities that at this resolution their EPI could also 'fit' (i.e. V530/I960, also at trimer interface (ie L890)). I think a good way to confirm this would be to do a focused Euler angle refinement on each protomer of chain separately (to get the best possible maps), then fit the ligand

at each possible site, then calculate the Q-score for each site. This would then add more validation that their modelled site is the 'most' correct one.

4. I think its good practice to also report the map-model FSC. I calculated it and it looks fine, but this is something that should probably go in the supplemental material.

5. Finally, I note that the protein is not well centred in the box. I can see why as in their reported 2D classes there is poor density for something on the intracellular face of the complex, and leading to cryoSPARC putting the resolved portion of the map not in the centre. This can lead to poorer resolution (especially of the inner membrane portion of the complex). It may be worthwhile recentring the map and re-refining to assess whether this helps the quality of the map.

6. Finally, this is more of a tip for next time. I note that the authors collected a very wide defocus range for their data collection. This would have limited their resolution. A range of only 0.1 um will fully fill all the missing zeros in the CTF and collecting between 0.5-0.8 micron defocus allows the higher spatial frequencies to be captured before they get washed out in Poisson noise.

This paper should be accepted.

Matthew Belousoff, Monash Institute of Pharmaceutical Sciences.

Referee #3 (Comments on Novelty/Model System for Author):

Overall, the paper is very well written (small, minor, corrections will help the non-specialist reader, see below), the strategy and workflow are convincing and the data are compelling. We have qualified the Ms with « medium » novelty because this paper is a follow up of a paper where the real proof of concept was made for *E. coli* (Plé Nature Communications 2022). As such, the study is hence not of « High » novelty as it is an adaptation to a new bacterium, *K. pneumoniae*. However, considering the critical nature of the latter bug, we believe that this study is of prime importance.

Referee #3 (Remarks for Author):

« Pyridylpiperazine efflux pump inhibitor boosts in vivo antibiotic efficacy against *K. pneumoniae* » by Anais Vieira Da Cruz et al. describes the characterization of new molecules optimised from a new class of inhibitors discovered by the authors and that were initially described to be efficient against the AcrAB-TolC efflux pump from *E. coli*. In the present study they rationally optimize the compounds by chemical synthesis in order to potentiate their activity towards *Klebsiella pneumoniae*.

One of such compounds, BDM 91288, is shown to bear physico-chemical and pharmacokinetic properties efficient enough to increase availability at the site of infection for in vivo efficacy studies.

Single-particle cryo-EM structural studies show that BDM91288 binds to a key region within the transmembrane segments of AcrB, where the energy for the efflux of drugs is known to be mediated. Finally, proof-of-concept studies demonstrate that oral administration of BDM91288 potentiates the in vivo efficacy of the antibiotic levofloxacin in a murine model of *K. pneumoniae* lung infection.

Overall, the paper is very well written, the strategy and workflow are convincing and the data are compelling. Beyond the minor corrections below, our main comment is that the authors have not proceeded to measurements of direct inhibition of the BDM compounds as was previously performed in their initial paper (Plé Nat Comm 2022), either regarding the bacterial accumulation of AcrB substrates (e.g berberine) or on the measurement of the potential stabilization of the inhibitor/AcrB complex by thermal short assay. Could the authors comment why such validations were not performed in the present study ?

Minor comments :

Page 5 :

« CLint = 59 µL/min/mg proteins » is said to be suboptimal for in vivo studies.

=> what would be the standard threshold ?

Page 6 :

« newly synthesised PyrPip analogues were initially evaluated on *E. coli* for their ability to boost pyridomycin antibiotic activity »

=> Why not directly on *Klebsiella* ?

Page 8, Table 1 :

Some metabolic microsomal stability values were not determined (« ND »). Why were they not ? Because not detectable ? Because of technological issues (that would have to be described).

Page 10, Table 2 :

Correct « PaβN » to « PABN ».

Page 14 :

« Furthermore, no toxicity was observed in mice after a single oral administration of BDM91288 (5) at 30 mg/kg ».

=> Was the absence of toxicity monitored over prolonged time period ? If so, how long ?

=> Was the mice strain used identical to that in PK studies ? If not, why ?

The data showed BDM91288 (5) to be bioavailable with detectable concentrations in the plasma ($C_{max} = 2.0 \mu\text{M}$, $T_{max} = 4 \text{ h}$, area under the curve ($\text{AUC}(0-24\text{h}) = 27 \text{ h}\cdot\mu\text{M}$), though below [...]

=> Please define AUC (area under curve) already here. In the present manuscript, AUC is defined page 15.

Page 15 :

« antibiotic activity is driven by the area under the curve (AUC)/MIC ratio (Scaglione et al, 2003) »

=> This sentence is quite obscure ... please rephrase.

Please define « p.o » and « i.p ».

Page 18 :

What does « phenocopy » means ?

« In addition, the activity of levofloxacin was boosted a maximum of 4-fold in vitro, suggesting that the use of another antibiotic, better substrate of the AcrAB-TolC pump, could lead to greater efficacy in vivo in combination with a PyrPip compound. A comprehensive PK/PD analysis is thus needed to find a better combination »

=> What substrate do the authors have in mind ?

Page 22 :

At what temperature is DDM solubilization performed ?

Supplementary Figure S1 :

Please number the AcrB TM segments as in Figure 1 (page 11).

Response to reviewer comments

Manuscript Number: EMM-2023-18236

Pyridylpiperazine efflux pump inhibitor boosts *in vivo* antibiotic efficacy against *K. pneumoniae*

First of all, the authors would like to thank the reviewers for their thoughtful comments and efforts towards improving the quality of the manuscript.

The responses to the points raised by the reviewers are given below using a different text-color type.

Referee #1 (Remarks for Author):

This study is focused on a preliminary characterization of an efflux pump inhibitor of a pyridylpiperazine class of molecules. These inhibitors were previously reported. Here, the authors carried out focused medicinal chemistry efforts to improve the metabolic stability of the previously reported hit, to evaluate its resistance potential and limited PK and *in vivo* studies. The compound 5 showed improved stability and an accumulation in lungs, but no improvement in the efflux inhibitory activity. The resistance markers were linked to a point mutation in the target AcrB and its regulatory system RamRA. The cryo-EM structure of the target AcrB from *K. pneumoniae* bound to 5 further validated the result. The compound potentiated a fluoroquinolone antibiotic levofloxacin in the *K. pneumoniae* pulmonary infection model.

Overall, studies are well-designed and carried out, the results are convincing and demonstrate that this class of compounds has a desired effect on target.

We thank the reviewer for these comments.

On the other hand, the study has also showed that compounds have a resistance liability that will limit their further development for clinical use.

We agree with the reviewer that the frequency of resistance to any antimicrobial therapy is an important factor to take into consideration in drug development, and is a factor that affects the majority of antibiotics. In the case of pyridylpiperazines and *K. pneumoniae*, we found it incredibly challenging to identify mutations in AcrB, which is in line with our research in *E. coli* where only a frequency of resistance of $\sim 1.5 \times 10^9$ was reported, a frequency that is relatively low for antibiotics. Non-AcrB mutations such as those identified in *RamR* can also decrease pyridylpiperazine activity and are likely more frequent (FOR not measured in this study), but as shown in the paper, the resulting AcrB overexpression can be overcome by higher pyridylpiperazine concentrations.

Another weakness is that the demonstrated activity is in a combination with levofloxacin. However, in clinics, the primary reason for fluoroquinolone resistance are mutations in topoisomerases, the target of these antibiotics. The reported compounds will have no clinical utility if further developed in a combination with levofloxacin.

In this study, we chose levofloxacin as a model antibiotic to demonstrate *in vivo* proof of concept. We are aware of the existence of bacterial strains with resistance to fluoroquinolones through mutations in topoisomerases and that our PyrPip compounds will not be able to boost levofloxacin in these strains. However, the advantage of developing AcrB inhibitors is that they will be active in combination with a wide range of existing but also future antibiotics. It will

therefore be possible to identify therapeutic alternatives for overcoming resistance in the mechanism of action of an antibiotic by using a combination of an EPI with another antibiotic.

Referee #2 (Remarks for Author):

The manuscript "Pyridylpiperazine efflux pump inhibitor boosts in vivo antibiotic efficacy against *K. pneumoniae*" by Cruz and coworkers describes the medicinal chemistry optimization and characterization of a bacterial efflux pump inhibitor (EPI). The work covers both the synthesis of a range of pyridylpiperazine derivatives and they are tested against a variety of *K. pneumoniae* strains that have increased AcrAB/TolC expression in the presence of a range of clinically used antibiotics. Their lead compound in this study (BDM91288) was then interrogated by single particle cryoEM against the AcrB trimer and its structure activity relationship was determined. Furthermore, this compound was then tested in a murine model lung infection study where in combination with levofloxacin, it showed better bacterial clearance than the antibiotic alone.

In my opinion this manuscript is of excellent quality and is a well-conceived and well carried out study that is of great import in the further development of new antibacterial treatments. However, I do have some minor comments, mostly based on the cryoEM analysis. I also want to thank the authors in sharing their data.

1. There are some minor modelling problems, one example is in ChainA; Y467,F459,L875 are all in wrong density, which has led to some Ramachandran outliers. This is true across the whole structure, so it would be ideal to carefully go through and ensure no other errors.

We thank the reviewer for pointing this out – we have fixed the errors now in the revised version. We updated the further refined model in the PDB database, and we provide the updated validation report here for review.

2. In the published structure 7OUK had piperazine compound in two positions. I note that there is weak density in Chain C (W808) (in the authors structure) for second allosteric site. Perhaps some focussed refinements on this area might also elucidate this second site.

We appreciate the reviewer's observation regarding the weak density in Chain C (W808) within our structure and the suggestion provided. Even after focused refinements in this specific region a density that can accommodate a second allosteric site for BDM91288 could not be resolved in our cryoEM structure.

In the 7OUK crystal structure a different compound (BDM88855) was used, which is smaller than our current compound (BDM91288). The structural geometry of BDM91288 with the second piperazine ring cannot be accommodated within the weak density in Chain C. Attempts to fit it led to massive clashes as shown in figure 1RA.

Figure 1R: **A.** Shown is the overlay of the potential placement of BDM91288 (light purple) with the binding site of BDM88855 (yellow) from 7OUK in our KpAcrB 8P1I structure (green). The respective cryoEM density map is shown in blue. The resulting clashes (regarding the position of BDM91288 in 8P1I) are shown in magenta. **B.** Shown is the binding position of BDM88855 (dark blue) in the 7OUK structure (dark blue) that is not leading to significant clashes in contrast to shown in A. The overlay of 8P1I (green) and the respective cryoEM density map (blue) indicates the altered conformation of the 7OUK structure in this specific region. The software WinCoot 0.9.8.7 was used for image generation.

3. While I completely agree with the authors placement of BDM91288 in ChainA, there are lots of other unaccounted for density (most of which appears to be various lipids, or detergent molecules). However, there are some other densities that at this resolution their EPI could also 'fit' (i.e. V530/I960, also at trimer interface (ie L890)). I think a good way to confirm this would be to do a focused Euler angle refinement on each protomer of chain separately (to get the best possible maps), then fit the ligand at each possible site, then calculate the Q-score for each site. This would then add more validation that their modelled site is the 'most' correct one.

We appreciate the reviewer's agreement with our placement of BDM91288 in Chain A. We thank the reviewer for the detailed feedback and agree that there are other unaccounted densities in the structure, however these densities are outside the core of the protomers (figure 2RA). As displayed in the illustration, the densities in green are located outside of the protein core (shown in orange), belonging to the DDM micelle or potentially being membrane lipids. The densities mentioned by the reviewer shown in blue at trimer interface (L890) are also outside the core of the AcrB protomers. The densities, which we could account for lipids with high enough confidence, were already incorporated into our structural model.

Two densities can be determined within the core of the protein. The first corresponds to the position of BDM91288 (neon-green, circle) in 8P1I, which we have identified by means of the already known position of BDM88855 from the published crystal structure 7OUK in Plé *et al.* The second density (magenta) aligns with the density proposed by the reviewer at position V530/I960, which could potentially accommodate the EPI. However, the tubular shape of the density at this position is to our experience rather reminiscent to an alkyl chain, such as we see for lipids or detergent. The EPI density which we assigned to BDM91288 has a more planar appearance (broader and more flat in three dimensions). In figure 2RB and C, we show that we cannot unambiguously fit BDM91288 into this density, but we also cannot exclude the possibility that this density accommodates the EPI. Unfortunately, focused Euler angle refinements on the individual protomer did not result in further improvement of this site.

Figure 2R. **A.** Overview of all densities (green, blue, magenta) that are not part of the KpAcrB protein structure (molmap, orange) in bottom view. **B.** Position of BDM91288 (neon-green) in 8P1I and potential placement of BDM91288 in the unaccounted density near V530/I960 (magenta). **C.** Potential placement of BDM91288 in the density near V530/I960 shown in the context of the protein density map in WinCoot 0.9.8.7.

4. I think its good practice to also report the map-model FSC. I calculated it and it looks fine, but this is something that should probably go in the supplemental material.

Thanks, we have included this information in the revised version.

5. Finally, I note that the protein is not well centred in the box. I can see why as in their reported 2D classes there is poor density for something on the intracellular face of the complex, and leading to cryoSPARC putting the resolved portion of the map not in the centre. This can lead to poorer resolution (especially of the inner membrane portion of the complex). It may be worthwhile recentring the map and re-refining to assess whether this helps the quality of the map.

We agree with the reviewer, the protein is often forming hexamers, which caused the slight shift within the box. However, we employed the local refinement job in cryoSPARC as our final refinement step, with the fulcrum location specifically set to the mask center, and not to the box center. In this regard, the shift does not have an effect in the final resolution, thus we did not repeat the processing in this aspect.

6. Finally, this is more of a tip for next time. I note that the authors collected a very wide defocus range for their data collection. This would have limited their resolution. A range of only 0.1 um will fully fill all the missing zeros in the CTF and collecting between 0.5-0.8 micron defocus allows the higher spatial frequencies to be captured before they get washed out in Poisson noise.

We appreciate the reviewer's suggestion. We will consider the options suggested for our further data analysis.

This paper should be accepted.

Matthew Belousoff, Monash Institute of Pharmaceutical Sciences.

Referee #3 (Comments on Novelty/Model System for Author):

Overall, the paper is very well written (small, minor, corrections will help the non-specialist reader, see below), the strategy and workflow are convincing and the data are compelling. We have qualified the Ms with « medium » novelty because this paper is a follow up of a paper where the real proof of concept was made for *E. coli* (Plé Nature Communications 2022). As such, the study is hence not of « High » novelty as it is an adaptation to a new bacterium, *K. pneumoniae*. However, considering the critical nature of the latter bug, we believe that this study is of prime importance.

Referee #3 (Remarks for Author):

« Pyridylpiperazine efflux pump inhibitor boosts in vivo antibiotic efficacy against *K. pneumoniae* » by Anais Vieira Da Cruz et al. describes the characterization of new molecules optimised from a new class of inhibitors discovered by the authors and that were initially described to be efficient against the AcrAB-TolC efflux pump from *E. coli*. In the present study they rationally optimize the compounds by chemical synthesis in order to potentiate their activity towards *Klebsiella pneumoniae*.

One of such compounds, BDM 91288, is shown to bear physico-chemical and pharmacokinetic properties efficient enough to increase availability at the site of infection for in vivo efficacy studies.

Single-particle cryo-EM structural studies show that BDM91288 binds to a key region within the transmembrane segments of AcrB, where the energy for the efflux of drugs is known to be mediated. Finally, proof-of-concept studies demonstrate that oral administration of BDM91288 potentiates the in vivo efficacy of the antibiotic levofloxacin in a murine model of *K. pneumoniae* lung infection.

Overall, the paper is very well written, the strategy and workflow are convincing and the data are compelling.

Beyond the minor corrections below, our main comment is that the authors have not proceeded to measurements of direct inhibition of the BDM compounds as was previously performed in their initial paper (Plé Nat Comm 2022), either regarding the bacterial accumulation of AcrB substrates (e.g berberine) or on the measurement of the potential stabilization of the inhibitor/AcrB complex by thermal shift assay. Could the authors comment why such validations were not performed in the present study?

We thank the reviewer for the kind comments and the question on the biophysical characterization of this *K. pneumoniae* AcrB BDM-inhibitor complex.

Indeed, for the characterization of pyridylpiperazines on *E. coli* AcrB (Plé et al., 2022) both TSA and berberine uptake studies were performed to complement the mechanism of action studies. These assays were not done in the submitted manuscript as we considered that the focus of this manuscript was on the pharmacokinetics and *in vivo* efficacy studies. We considered that the mechanism of action was important and conducted the cryo-EM studies to assess the binding of BDM 91288 to *KpAcrB*. Nonetheless, after your comment, we considered that the berberine uptake assay can provide data supporting that pyridylpiperazines are able to act immediately on efflux, rather than indirectly through, for example, *acrB* expression. Therefore, we followed your suggestion to exclude this possibility and we analysed the impact of BDM91288 on berberine uptake in both WT *K. pneumoniae* and the AcrB (V448E) variant and could confirm that berberine uptake was inhibited only in the *K. pneumoniae* strain harbouring the AcrB wildtype and not with the strain harbouring the V448E variant. The apparent K_i value in this assay and under these conditions was determined to be around 20 μM .

The following text and figure were added to the manuscript:

Direct inhibition of *K. pneumoniae* AcrB by BDM91288 (5)

To confirm the direct inhibition of AcrB by PyrPips, we assessed the impact of BDM91288 (5) on the immediate accumulation of the AcrB substrate berberine in *K. pneumoniae* using a fluorescence uptake assay. As expected, BDM91288 (5) mediated a concentration-dependent stimulation of berberine uptake in *K. pneumoniae* WT with an apparent K_i of around 20 μM , which was not observed for *K. pneumoniae* producing the AcrB variant V448E (Fig EV1).

Figure EV1 - Rate of berberine uptake by *K. pneumoniae* WT and *K. pneumoniae* AcrB (V448E) in the presence of a gradient of BDM91288 (5) concentrations.

Data are a mean and SD of 4-independent experiments.

Minor comments :

Page 5 :

« $CL_{int} = 59 \mu\text{L}/\text{min}/\text{mg proteins}$ » is said to be suboptimal for *in vivo* studies.
=> what would be the standard threshold ?

Usually $CL_{int} < 20 \mu\text{L}/\text{min}/\text{mg proteins}$ is considered suitable for an *in vivo* pharmacokinetic experiment (Houston, J.B. Utility of *in vitro* drug metabolism data in predicting *in vivo* metabolic clearance. *Biochemical Pharmacology*, 1994, 47, 1469-1479. doi.org/10.1016/0006-2952(94)90520-7). The sentence has been amended as follows: "For this reason, we sought to further explore the structure activity relationships of PyrPips to identify inhibitors with improved metabolic stability (**$CL_{int} < 20 \mu\text{L}/\text{min}/\text{mg proteins}$**)."

Page 6 :

« newly synthesised PyrPip analogues were initially evaluated on *E. coli* for their ability to boost pyridomycin antibiotic activity »

=> Why not directly on *Klebsiella* ?

The compounds synthesised are first tested on a single type of Enterobacteriales, and we have chosen *E. coli*. Interesting compounds were then tested on *K. pneumoniae*. Given the high similarity between the *E. coli* and *K. pneumoniae* AcrB proteins, we observed the same boosting effect on both bacteria.

Page 8, Table 1 :

Some metabolic microsomal stability values were not determined (« ND »). Why were they not ? Because not detectable ? Because of technological issues (that would have to be described).

Only the microsomal stability of the most potent compounds ($EC_{90} < 4 \mu\text{M}$) was measured, as this experiment is time and cost consuming. We also measured the microsomal stability of compounds 6 and 7, which are less potent, to see the advantage of introducing a piperazine (compound 5) on microsomal stability over a morpholine (compound 6) or a piperidine (compound 7).

The following sentence has been added in the text: "The microsomal stability of the most potent compounds ($EC_{90} < 4 \mu\text{M}$) was then measured."

Page 10, Table 2 :

Correct « PaßN » to « PAßN ».

The text has been corrected.

Page 14 :

« Furthermore, no toxicity was observed in mice after a single oral administration of BDM91288 (5) at 30 mg/kg ».

=> Was the absence of toxicity monitored over prolonged time period ? If so, how long ?

The absence of toxicity was observed over a 24-hour period which corresponds to the final time point in PK and efficacy experiments.

The sentence has been amended as follows: "Furthermore, no toxicity was observed in mice **over a 24-hour period** after a single oral administration of BDM91288 (5) at 30 mg/kg".

=> Was the mice strain used identical to that in PK studies ? If not, why ?

Female CD-1 mice were used for the PK experiments while female C57BL/6JRj mice were used for efficacy studies. Mice used for each experiment is indicated in the text, in figure 3 and in

Materials and Methods section. We used CD-1 mice for PK experiments because they are much cheaper than C57BL/6JRj mice.

In the publication of John T. Barr *et al.* entitled "Strain-Dependent Variability of Early Discovery Small Molecule Pharmacokinetics in Mice: Does Strain Matter?" (Drug Metab Dispos, 2020, 48(8), 613-621, doi: 10.1124/dmd.120.090621) the authors wrote "The mouse strain in discovery PK studies may not match the strain in efficacy and tox studies. Currently, there is gap in the literature addressing if differences in PK parameters across mouse strains exist such that multiple PK studies are warranted. The results from this study indicated that the PK properties of clinically used drugs between mouse strains are within an acceptable range such that single strain PK is appropriate."

The following sentence has been added page 21: Two different mice strains were used for PK (CD-1) and efficacy (C57BL/6JRj) studies as it was shown in the literature that there was little difference in the PK parameters measured between mice strains.

The data showed BDM91288 (5) to be bioavailable with detectable concentrations in the plasma ($C_{max} = 2.0 \mu\text{M}$, $T_{max} = 4 \text{ h}$, area under the curve (AUC(0-24h)) = 27 h. μM), though below [...]

=> Please define AUC (area under curve) already here. In the present manuscript, AUC is defined page 15.

The sentence has been amended as follows: "The data showed BDM91288 (5) to be bioavailable with detectable concentrations in the plasma ($C_{max} = 2.0 \mu\text{M}$, $T_{max} = 4 \text{ h}$, **area under the plasma drug concentration-time curve** (AUC(0-24h)) = 27 h. μM).

Page 15 :

« antibiotic activity is driven by the area under the curve (AUC)/MIC ratio (Scaglione et al, 2003) »

=> This sentence is quite obscure ... please rephrase.

This sentence has been replaced by "In this work levofloxacin was chosen as an appropriate partner antibiotic because the ratio of the pharmacokinetic area under the plasma concentration-time curve (AUC) to the MIC (i.e., AUC/MIC) best correlates with its *in vivo* efficacy (Scaglione et al, 2003). Therefore, by decreasing its MIC 4-fold on *K. pneumoniae* ATCC 43816 with 38 μM of BDM91288 (5) (Fig 2) we expect an increase in the AUC/MIC ratio and a boost of levofloxacin efficacy when combined with this PyrPip *in vivo*."

Please define « p.o » and « i.p ».

These abbreviations have been defined. The sentence page 14 has been amended as follows: "PK studies in female mice (CD-1) following administration of a single BDM91288 (5) dose (30 mg/kg, **per os** (p.o.)) with a single levofloxacin dose (10 mg/kg, **intraperitoneal** (i.p.))"

Page 18 :

What does « phenocopy » means ?

The sentence "However, the boosting of 10 mg/kg of levofloxacin by 30 mg/kg BDM91288 (5) could not phenocopy the efficacy of 50 mg/kg levofloxacin suggesting incomplete inhibition of AcrB in this set up" has been amended as follows: "However, the combination of 10 mg/kg of levofloxacin with 30 mg/kg BDM91288 (5) was not as effective as 50 mg/kg levofloxacin suggesting incomplete inhibition of AcrB in this set up".

« In addition, the activity of levofloxacin was boosted a maximum of 4-fold *in vitro*, suggesting that the use of another antibiotic, better substrate of the AcrAB-TolC pump, could lead to greater efficacy *in vivo* in combination with a PyrPip compound. A comprehensive PK/PD analysis is thus needed to find a better combination »

=> What substrate do the authors have in mind ?

For example, the activity of azithromycin can be boosted at least 8-fold in combination with EPIs (Table 2), so this antibiotic could lead to greater efficacy *in vivo*.

Page 22 :

At what temperature is DDM solubilization performed ?

The solubilisation was performed at 4 °C. This information has been added in the M&M.

Supplementary Figure S1 :

Please number the AcrB TM segments as in Figure 1 (page 11).

We introduced the TM numbers in the revised figure.

2nd Nov 2023

Dear Dr. Flipo,

Thank you for the submission of your revised manuscript to EMBO Molecular Medicine. We have now received the enclosed reports from the referees that were asked to re-assess it. As you will see the reviewers are now globally supportive and I am pleased to inform you that we will be able to accept your manuscript pending the following final amendments:

- 1) We note that you currently have together with you, a total of 4 co-corresponding authors. Is that correct? Do you confirm equal contribution of these 4 people, able to take full responsibility for the paper and its content? While there is no limit per se to the number of co-corresponding authors, 4 is rare, and may not reflect as intended to the community.
- 2) In the main manuscript file, please do the following:
 - Data availability: Please be aware that all deposited datasets should be freely accessible upon publication. Also, please ensure that a URL or DOI is included for your sequencing and structure datasets using the following as a guideline: [data type]: [full name of the resource] [accession number/identifier] ([doi or URL or identifiers.org/DATABASE:ACCESSION])
 - The sentence on the use of BioRender should be removed from the Acknowledgements and added to the Data Availability statement instead using the following format:
Graphics: (some of the... OR Figure #... OR synopsis) Graphics were created with BioRender.com.
 - Materials and Methods: Please double check the first sentence of the Materials and Methods - it appears that the structural validation of the compounds can be found in the Appendix rather than the Source Data.
 - Please rename "Competing interests statement" to "Disclosure and competing interests statement". We updated our journal's competing interests policy in January 2022 and request authors to consider both actual and perceived competing interests. Please review the policy <https://www.embopress.org/competing-interests> and update your competing interests if necessary.
- 3) Tables: Table 1 should be provided in a format that can be typeset (no image file). Alternatively, upload as a separate file (with legend) and rename "Table EV1".
- 4) The Paper Explained text: Please check whether the following sentence should be edited ('antibiotic' or 'antibiotics?') "This threat has led the World Health Organisation to prioritise research and development of antibiotic for drug-resistant Enterobacterales such as Escherichia coli and Klebsiella pneumoniae."
- 5) Synopsis:
 - Synopsis image: The Synopsis image needed to be resized (please find resized version attached). Please check whether you are okay with the size of the text and graphics in the resized version or consider simplifying.
 - Please check your synopsis text and image before submission with your revised manuscript. Please be aware that in the proof stage minor corrections only are allowed (e.g., typos).
- 6) For more information: This space should be used to list relevant web links for further consultation by our readers. Could you identify some relevant ones and provide such information as well? Some examples are patient associations, relevant databases, OMIM/proteins/genes links, author's websites, etc...
- 7) Appendix file: Please add a title to the beginning of the list of compounds starting on page 19.
- 8) Funding: Please ensure the complete list of funders listed in the Acknowledgements is also given in our manuscript submission system (no funding IDs or project numbers required).
- 9) As part of the EMBO Publications transparent editorial process initiative (see our Editorial at <http://embomolmed.embopress.org/content/2/9/329>), EMBO Molecular Medicine will publish online a Review Process File (RPF) to accompany accepted manuscripts. This file will be published in conjunction with your paper and will include the anonymous referee reports, your point-by-point response and all pertinent correspondence relating to the manuscript. Let us know whether you agree with the publication of the RPF and as here, if you want to remove or not any figures from it prior to publication. Please note that the Authors checklist will be published at the end of the RPF.
- 10) Please provide a point-by-point letter INCLUDING my comments as well as the reviewer's reports and your detailed responses (as Word file).

I look forward to reading a new revised version of your manuscript as soon as possible.

Yours sincerely,

Poonam Bheda

Poonam Bheda, PhD
Scientific Editor
EMBO Molecular Medicine

**** Reviewer's comments ****

Referee #2 (Remarks for Author):

I am satisfied that the authors made every effort to address the reviewers concerns and comments it would be my recommendation for the manuscript to be published.

Referee #3 (Comments on Novelty/Model System for Author):

This new version tackles the corrections and issues raised by the reviewers. The overall scope and message is the same as the initial version so I keep my initial appreciation on the latter topics.

Referee #3 (Remarks for Author):

The authors have satisfactorily addressed my comments and suggestions. The manuscript is very much improved and is, in my opinion, suitable for publication.

The authors addressed the minor editorial issues.

17th Nov 2023

Dear Dr. Flipo,

Congratulations on your very nice work. We are pleased to inform you that your manuscript is accepted for publication and is now being sent to our publisher to be included in the next available issue of EMBO Molecular Medicine.

Yours sincerely,

Poonam Bheda, PhD
Scientific Editor
EMBO Molecular Medicine